**1** **Estimating the snow water equivalent on a glacierized high elevation site (Forni Glacier, Italy)**

**3** Senese Antonella[1], Maugeri Maurizio[1], Meraldi Eraldo[2], Verza Gian Pietro[3], Azzoni Roberto Sergio[1],

**4** Compostella Chiara[4], Diolaiuti Guglielmina[1]

**6** [1] Department of Environmental Science and Policy, Università degli Studi di Milano, Milan, Italy.

**7** [2] ARPA Lombardia, Centro Nivometeorologico di Bormio, Bormio, Italy.

**8** [3] Ev-K2-CNR - Pakistan, Italian K2 Museum Skardu Gilgit Baltistan, Islamabad, Pakistan.

**9** [4] Department of Earth Sciences, Università degli Studi di Milano, Milan, Italy.

**11** *Correspondence to*: Antonella Senese (antonella.senese@unimi.it)

**13** **Abstract.**

**14** We present and compare 11 years of snow data (snow depth and snow water equivalent, *SWE*) measured by an Automatic

**15** Weather Station and corroborated by data from field campaigns on the Forni Glacier in Italy. The aim of the analysis is to

**16** estimate the *SWE* of new snowfall and the annual *SWE* peak based on the average density of the new snow at the site (corre-

**17** sponding to the snowfall during the standard observation period of 24 hours) and automated snow depth measurements. The

**18** results indicate that the daily SR50 sonic ranger measures and the available snow pit data can be used to estimate the mean

**19** new snow density value at the site, with an error of $\pm 6$ kg m$^{-3}$. Once the new snow density is known, the sonic ranger makes

**20** it possible to derive *SWE* values with a RMSE of 45 mm water equivalent (if compared with snow pillow measurements),

**21** which turns out to be about 8% of the total *SWE* yearly average. Therefore, the methodology we present is interesting for

**22** remote locations such as glaciers or high alpine regions, as it makes it possible to estimate the total snow water equivalent

**23** (*SWE*) using a relatively inexpensive, low power, low maintenance, and reliable instrument such as the sonic ranger.

**24**

**25**

**26** **Keywords**: Snow depth; Snow water equivalent (SWE); SPICE (Solid Precipitation Intercomparison Experiment) project;

**27** Forni Glacier.

**28**

**29**

**30** **1. Introduction and scientific background**

**31** The study of the spatial and temporal variability of water resources deriving from snow melt (i.e. Snow Water Equivalent,

**32** *SWE*) is very important for estimating the water balance at the catchment scale. Many areas depend on this freshwater reservoir

**33** for civil use, irrigation and hydropower, so they need an accurate and updated evaluation of *SWE* magnitude and variability.

**34** In addition, a correct *SWE* assessment also supports early strategies for managing and preventing hydro-meteorological risks

(e.g. flood forecasting, avalanche forecasting). New snow-density evaluation is also important for snowfall forecasting based
on orographic precipitation models (Judson and Doesken, 2000; Roebber et al., 2003), estimation of avalanche hazards (Perla,
1970; LaChapelle, 1980; Ferguson et al., 1990; McClung and Schaerer, 1993), snowdrift forecasting, and as an input param-
eter in the snow accumulation algorithm (Super and Holroyd, 1997), and general snow science research.
In high mountain areas, however, often only snowfall measures are available: a correct evaluation of new snow density
($\rho_{new\ snow}$) is therefore needed to calculate the *SWE*. Since new snow density is site specific and depends on atmospheric and
surface conditions, the main aim of this study is to investigate the magnitude and rates of variations in $\rho_{new\ snow}$ and to under-
stand how an incorrect assessment of this variable may affect the estimation of the *SWE*. This was possible by means of
systematic manual and automatic measurements carried out at the surface of the Forni Glacier (Stelvio Park, Italian Alps, Fig.
1a and b). Since 2005, an Automatic Weather Station (AWS1 Forni) has been acquiring snow data at the glacier surface, in
addition to snow pit measurements of snow depth and *SWE* carried out by expert personnel (Citterio et al., 2007; Senese et
al., 2012a; 2012b; 2014). The snow data thus acquired refer to snowfall or new snow (i.e. depth of freshly fallen snow depos-
ited over a standard observation period, generally 24 hours, see WMO, 2008; Fierz et al., 2009) and to snow depth (i.e. the
total depth of snow on the ground at the time of observation, see WMO, 2008).
In general, precipitation can be measured mechanically, optically, by capacitive sensing and by radar. Some examples of
available sensors are: the heated tipping bucket rain gauge (as precipitation is collected and melted in the gauge's funnel,
water is directed to a tipping bucket mechanism adjusted to tip and dump when a threshold volume of water is collected), the
heated weighing gauge (the weight of water collected is measured as a function of time and converted to rainfall depth), and
the disdrometer (measuring the drop size distribution and the velocity of falling hydrometeors). For catchment type precipi-
tation sensors, the catch efficiency of solid precipitation needs to be considered for the correct measurement of new snow.
For the Solid Precipitation Intercomparison Experiment (1989-1993), the International Organizing Committeee designated
the Double Fence Intercomparison Reference (DFIR) as the reference for intercomparison (WMO/TD-872/1998, section
2.2.2). Even if all these methods mentioned provide accurate measurements, it is very difficult to utilize some of them in
remote areas like a glacier site. For this reason, at the Forni Glacier, snow data have been acquired by means of sonic ranger
and snow pillow instrumentations, without wind shielding.
For estimating *SWE* from snow depth measurements alone, a correct new snow density estimate is crucial. Following Roebber
et al. (2003), new snow density is often assumed to conform to the 10-to-1 rule: the snow ratio, defined by the density of
water (1000 kg m$^{-3}$) to the density of new snow (assumed to be 100 kg m$^{-3}$), is 10:1. As noted by Judson and Doesken (2000),
the 10-to-1 rule appears to originate from the results of a nineteenth-century Canadian study. More comprehensive measure-
ments (e.g., Currie, 1947; LaChapelle, 1962; Power et al., 1964; Super and Holroyd, 1997; Judson and Doesken, 2000) have
established that this rule is an inadequate characterization of the true range of new snow densities. Indeed, they can vary from
10 kg m$^{-3}$ to approximately 350 kg m$^{-3}$ (Roebber et al., 2003). Bocchiola and Rosso (2007) report a similar range for the
Central Italian Alps with values varying from 30 kg m$^{-3}$ to 480 kg m$^{-3}$, and an average sample value of 123 kg m$^{-3}$. The lower
bound of new snow density is usually about 50 kg m$^{-3}$ (Gray, 1979; Anderson and Crawford, 1990). Judson and Doesken
(2000) found densities of new snow observed from six sheltered avalanche sites in the Central Rocky Mountains to range
from 10 to 257 kg m$^{-3}$, and average densities at each site based on four years of daily observations ranged from 72 to 103 kg
m$^{-3}$. Roebber et al. (2003) found that the 10-to-1 rule may be modified slightly to 12 to 1 or doubled to 20 to 1, depending on
the mean or median climatological value of new snow density at a particular station (e.g. Currie 1947; Super and Holroyd,
1997). Following Pahaut (1975), the new snow density ranges from 20 to 200 kg m$^{-3}$ and increases with wind speed and air
temperature. Wetzel and Martin (2001) analyzed all empirical techniques evolved in the absence of explicit snow-density
forecasts. As argued in Schultz et al. (2002), however, these techniques might be not fully adequate and the accuracy should
be carefully verified for a large variety of events.
New snow density is regulated by i) in-cloud processes that affect the shape and size of ice crystal growth, ii) sub-cloud
thermodynamic stratification through which the ice crystals fall (since the low-level air temperature and relative humidity
regulate the processes of sublimation or melting of a snowflake), and iii) ground-level compaction due to prevailing weather
conditions and snowpack metamorphism. Understanding how these processes affect new snow density is difficult because
direct observations of cloud microphysical processes, thermodynamic profiles, and surface measurements are often unavail-
able.
Cloud microphysical research indicates that many factors contribute to the final structure of an ice crystal. The shape of the
ice crystal is determined by the environment in which the ice crystal grows: pure dendrites have the lowest density (Power et
al., 1964), although the variation in the density of dendritic aggregates is large (from approximately 5 to 100 kg m$^{-3}$, Magono
and Nakamura, 1965; Passarelli and Srivastava, 1979). Numerous observational studies over decades clearly demonstrate that
the density varies inversely with size (Magono and Nakamura, 1965; Holroyd, 1971; Muramoto et al., 1995; Fabry and Szyr-
mer, 1999; Heymsfield et al., 2004; Brandes et al., 2007). The crystal size is related to the ratio between ice and air (Roebber
et al., 2003): large dendritic crystals will occupy much empty air space, whereas smaller crystals will pack together into a
denser assemblage. In addition, as an ice crystal falls, it passes through varying thermodynamic and moisture conditions.
Then, the ultimate shape and size of crystals depend on factors that affect the growth rate and are a combination of various
growth modes (e.g. Pruppacher and Klett, 1997).
To contribute to the understanding of all the above topics, in this paper we discuss and compare all the available snow data
measured at the Forni Glacier surface in the last decade to: i) suggest the most suitable measurement system for evaluating
*SWE* at the glacier surface (i.e. snow pillow, sonic ranger, snow pit or snow weighing tube); ii) assess the capability to obtain
*SWE* values from the depth measurements and their accuracies; iii) check the validity of the $\rho_{new\ snow}$ value previously found
(i.e. 140 kg m$^{-3}$, see Senese et al., 2014) in order to support *SWE* computation; and iv) evaluate effects and impacts of uncer-
tainties in the $\rho_{new\ snow}$ value in relation to the derived *SWE* amount.


**2.  Study area and Forni AWSs**
The Forni Glacier (one of the largest glaciers in Italy) is a Site of Community Importance (SCI, code IT2040014) located
inside an extensive natural protected area (the Stelvio Park). It is a wide valley glacier (ca. 11.34 km$^2$ of area, D'Agata et al.,
2014), covering an elevation range from 2600 to 3670 m a.s.l..
The first Italian supraglacial station (AWS1 Forni, Fig. 1b) was installed on 26[th] September 2005 at the lower sector of the
eastern tongue of Forni Glacier (Citterio et al., 2007; Senese et al., 2012a, 2012b; 2014; 2016). The WGS84 coordinates of
AWS1 Forni were: 46° 23' 56.0" N, 10° 35' 25.2" E, 2631 m a.s.l. (Fig. 1a, yellow triangle). The second station (AWS Forni
SPICE, Fig. 1b) was installed on 6[th] May 2014 close to AWS1 Forni (at a distance of about 17 m). Due to the formation of
ring faults, in November 2015 both AWSs were moved to the Forni Glacier central tongue (46°23'42.40"N and 10°35'24.20"E
at an elevation of 2675 m a.s.l., the red star in Fig. 1a). Ring faults are a series of circular or semicircular fractures with

stepwise subsidence (caused by englacial or subglacial meltwater) that could compromise the stability of the stations because they could create voids at the ice-bedrock interface and eventually the collapse of cavity roofs (Azzoni et al., 2017; Fugazza et al., 2017).

The main challenges in installing and managing Forni AWSs were due to the fact that the site is located on the surface of an Alpine glacier, not always accessible, especially during wintertime when skis and skins are needed on the steep and narrow path, and avalanches can occur. Moreover, the glacier is a dynamic body (moving up to 20-30 m y$^{-1}$, Urbini et al., 2017) and its surface also features a well-developed roughness due to ice melting, flowing meltwater, differential ablation and opening crevasses (Diolaiuti and Smiraglia, 2010; Smiraglia and Diolaiuti, 2011). In addition, the power to be supplied to instruments and sensors is only provided by solar panels and lead-gel batteries. A thorough and accurate analysis of instruments and devices (i.e. energy supply required, performance and efficiency operation at low temperatures, noise in measuring due to ice flow, etc.) was required before their installation on the supraglacial AWSs to avoid interruptions in data acquisition and storage.

AWS1 Forni is equipped with sensors for measuring air temperature and humidity (a naturally ventilated shielded sensor), wind speed and direction, air pressure, and the four components of the radiation budget (longwave and shortwave, both incoming and outgoing fluxes). Liquid precipitation is measured by means of an unheated precipitation gauge, and snow depth by means of the Campbell SR50 sonic ranger (Table 1, see also Senese et al., 2012a).

AWS Forni SPICE is equipped with a snow pillow (Park Mechanical steel snow pillow, 150 x 120 x 1.5 cm) and a barometer (STS ATM.1ST) for measuring the snow water equivalent (Table 1, Beaumont, 1965). The measured air pressure permits calibration of the output values recorded by the snow pillow. The snow pillow pressure gauge is a device similar to a large air or water mattress filled with antifreeze. As snow is deposited on this gauge, the pressure increase is related to the accumulating mass and thus to *SWE*. On the mast, an automated camera was installed to photograph the four graduated stakes located at the corners of the snow pillow (Fig. 1b) in order to observe the snow depth. When the snow pillow was installed at AWS Forni SPICE, a second sonic ranger (Sommer USH8) was installed at AWS1 Forni.

The whole systems of both AWS1 Forni and AWS Forni SPICE are supported by four-leg stainless steel masts (5 m and 6 m high, respectively) standing on the ice surface. In this way, the AWSs stand freely on the ice, and move together with the melting surface during summer (with a mean ice thickness variation of about 4 m per year).

The automated instruments are sampled every 60 seconds. The SR50 sonic ranger, wind sensor and barometer samples are averaged every 60 minutes. The air temperature, relative humidity, solar and infrared radiation, and liquid precipitation sample are averaged every 30 minutes. The USH8 sonic ranger and snow pillow sample are averaged every 10 minutes. All data are recorded in a flash memory card, including the basic distribution parameters (minimum, mean, maximum, and standard deviation values).

The long sequence of meteorological and glaciological data permitted the introduction of the AWS1 Forni into the SPICE (Solid Precipitation Intercomparison Experiment) project managed and promoted by the WMO (World Meteorological Organization) (Nitu et al., 2012) and the CryoNet project (Global Cryosphere Watch's core project, promoted by the WMO) (Key et al., 2015).

## 3. Data and methods

Snow data at the Forni Glacier have been acquired by means of i) a Campbell SR50 sonic ranger from October 2005 (snow depth data), ii) manual snow pits from January 2006 (snow depth and *SWE* data), iii) a Sommer USH8 sonic ranger from May 2014 (snow depth data), iv) a Park Mechanical SS-6048 snow pillow from May 2014 (*SWE* data), v) a manual snow weighing tube (Enel-Valtecne ©) from May 2014 (snow depth and *SWE* data). These measurements were made at the two automatic weather stations (AWSs): AWS1 Forni and AWS Forni SPICE.

Comparing the datasets from the Campbell and Sommer sensors, a very good agreement is found (r = 0.93). This means that both sensors have worked correctly. In addition, from 2015 onwards, the double snow depth datasets could mean better data for the *SWE* estimate.

In addition to the measurements recorded by the AWSs, since winter 2005-2006, personnel from the Centro Nivo-Meteoro-logico (namely CNM Bormio-ARPA Lombardia) of the Lombardy Regional Agency for the Environment have periodically used snow pits (performed according to the AINEVA protocol, see also Senese et al., 2014) in order to estimate snow depth and *SWE* (in mm water equivalent, w.e.). In particular, for each snow pit *j*, the thickness ($h_{ij}$) and the density ($\rho_{ij}$) of each snow layer (*i*) are measured for determining its snow water equivalent, and then the total $SWE_{snow\text{-}pit\text{-}j}$ of the whole snow cover (*n* layers) is obtained:

$$SWE_{snow-pit-j} = \sum_{i=1}^{n} h_{ij} \cdot \frac{\rho_{ij}}{\rho_{water}} \tag{1}$$

where $\rho_{water}$ is water density. As noted in a previous study (Senese et al., 2014), the date when the snow pit is dug is very important for not underestimating the actual accumulation. For this reason, we considered only the snow pits excavated before the beginning of snow ablation. In fact, whenever ablation occurs, successive *SWE* values derived from snow pits show a decreasing trend (i.e., they are affected by mass losses).

The snow pit *SWE* data were then used, together with the corresponding total new snow derived from sonic ranger readings, to estimate the site average $\rho_{new\,snow}$, in order to update the value of 140 kg m$^{-3}$ that was found in a previous study of data of the same site covering the period 2005-2009 (Senese et al., 2012a). We need to update our figures for $\rho_{new\,snow}$ as this is the key variable for estimating *SWE* from the sonic ranger's new snow data. Specifically, for each snow pit *j*, the corresponding total new snow was first determined by:

$$\Delta h_{snow-pit-j} = \sum_{t=1}^{m}(\Delta h_{tj}) \tag{2}$$

where *m* is the total number of days with snowfall in the period corresponding to snow pit *j*, and $\Delta h_{tj}$ corresponds to the depth of new snow on day *t*. Indeed, the new snow is defined as the depth of freshly fallen snow deposited over a standard observation period, generally 24 hours (see WMO, 2008; Fierz et al., 2009). In particular, we considered the hourly snow depth values recorded by the sonic ranger in a day and we calculated the difference between the last and the first reading. Whenever this difference is positive (at least 1 cm), it corresponds to a new snowfall. All data are subject to a strict quality control to avoid under- or over-measurements, to remove outliers and nonsense values, and to filter possible noise. $\sum_{t=1}^{m}(\Delta h_{tj})$ is therefore the total new snow measured by the Campbell SR50 from the beginning of the accumulation period to the date of the snow pit survey. Obviously, this value is higher than the snow depth recorded by the sonic ranger when the snow pit is dug, due to settling.

The average site $\rho_{new\,snow}$ was then determined as:

$$\rho_{new\,snow} = \frac{\sum_{j=1}^{k} SWE_{snow-pit-j}}{\sum_{j=1}^{k}(\Delta h_{snow-pit-j})} \tag{3}$$

where $j$ identifies a given snow pit and the corresponding total new snow, and the sum extends over all $k$ available snow pits. Instead of a mere average of $\rho_{new\ snow}$ values obtained from individual snow pit surveys, this relation gives more weight to snow pits with a higher $SWE_{snow-pit}$ amount.

The $SWE_{SR}$ (from sonic ranger data) for each day ($t$) was then estimated by:

$$SWE_{SR-t} = \begin{cases} \Delta h_t \frac{\rho_{new\ snow}}{\rho_{water}} & if\ \Delta h_t \geq 1\ cm \\ 0 & if\ \Delta h_t < 1\ cm \end{cases} \tag{4}$$

## 4. Results

Figure 2 represents the 11-year dataset of snow depth measured by the SR50 sonic ranger from 2005 to 2016. The last data (after October 2015) were recorded in a different site than the previous one because of the AWS's relocation in November 2015. The distance between the two sites is about 500 m, the difference in elevation is only 44 m and the aspect is very similar, so we do not expect the site change to have a noticeable impact on the snow depth data. A large inter-annual variability is seen, with a peak of 280 cm (on 2$^{nd}$ May 2008). In general, the maximum snow depth exceeds 200 cm, except in the period 2006-2007, which is characterized by the lowest maximum value (134 cm on 26$^{th}$ March 2007). These values are in agreement with findings over the Italian Alps in the period 1960–2009. In fact, Valt and Cianfarra (2010) reported a mean snow depth of 233 cm (from 199 to 280 cm) for the stations above 1500 m a.s.l. The snow accumulation period generally starts in late September to early October. The snow appears to be completely melted between the second half of June and the beginning of July (Fig. 2).

Because of the incomplete dataset from the Sommer USH8 sonic ranger, only the data from the Campbell SR50 sensor are considered for analysis.

The updated value of $\rho_{new\ snow}$ is 149 kg m$^{-3}$, which is similar to findings considering the 2005-2009 dataset (equal to 140 kg m$^{-3}$, (Senese et al., 2012a). Figure 3 reports the cumulative $SWE_{SR}$ values (i.e. applying Eq. 4) and the ones obtained using snow pit techniques ($SWE_{snow-pit}$) from 2005 to 2016. As found in previous studies (Senese et al., 2012a, 2014), there is a rather good agreement (RMSE = 58 mm w.e. with a mean $SWE_{snow-pit}$ value of 609 mm w.e.) between the two datasets (i.e. measured $SWE_{snow-pit}$ and derived $SWE_{SR}$). Whenever sonic ranger data are not available for a long period, the derived total $SWE$ value appears to be incorrect. In particular, in addition to the length of the missing dataset, the period of the year with missing data influences the magnitude of the underestimation of the actual accumulation. During the snow accumulation period 2010-2011, the data gap from 15 December 2010 to 12 February 2011 (a total of 60 days) produces an underestimation of 124 mm w.e. corresponding to 16% of the measured value (on 25 April 2011 $SWE_{SR}$ = 646 mm w.e. and $SWE_{snow-pit}$ = 770 mm w.e., Fig. 3). During the hydrological years 2011-2012 and 2012-2013, there were some problems with sonic ranger data acquisition thus making it impossible to accumulate these data from 31 January 2012 to 25 April 2013. In these cases, there are noticeable differences between the two datasets: on 1 May 2012 $SWE_{snow-pit}$ = 615 mm w.e. and $SWE_{SR}$ = 254 mm w.e., and on 25 April 2013 $SWE_{snow-pit}$ = 778 mm w.e. and $SWE_{SR}$ = 327 mm w.e., with an underestimation of 59% and 58%, respectively (Fig. 3). Figure 4 reports the comparison between the $SWE_{SR}$ values and the ones obtained using the snow pillow (for the period 2014-2016). Apart from a first interval without snow cover, or with just a very thin layer, the $SWE_{SR}$ curve follows that of $SWE$ measured by the snow pillow (Fig. 4), thus suggesting that our approach seems to offer reasonable results. In order to better assess the reliability of our derived $SWE_{SR}$ values, a scatter plot of measured $SWE$ data (by means of snow pillow, snow

weighing tube and snow pit) versus derived is shown (Fig. 5). The period chosen is the snow accumulation time frame during 2014-2015 and 2015-2016: from November 2014 to March 2015 and from February 2016 to May 2016 (i.e. the snow accumulation period, excluding the initial period in which the snow pillow seems to have significant measuring problems). There is a general underestimation of $SWE_{SR}$ compared to the snow pillow values, considering the 2014-2015 data, though the agreement strengthens in the 2015-2016 dataset (Fig. 5): 54 mm w.e. and 29 mm w.e. of RMSE regarding 2014-2015 and 2015-2016, respectively. Considering the whole dataset, the RMSE is 45 mm w.e., which proves to be about 8% of the total $SWE$ yearly average, as measured by the snow pillow. If compared with the snow pit, the difference is 35 mm w.e. (about 6% of the measured value). Nevertheless, numerous measurements made using the snow weighing tube (Enel-Valtecne ©) around the AWSs on 20 February 2015, showed wide variations of snow depth over the area (mean value of 165 cm and standard deviation of 29 cm), even if the snow surface seemed to be homogenous. This was mainly due to the roughness of the glacier ice surface. Indeed, on the same date, the snow pillow recorded a $SWE$ value of 493 mm w.e., while from the snow pit the $SWE$ was equal to 555 mm w.e., and from the snow weighing tube the $SWE$ ranged from 410 to 552 mm w.e. (Fig. 5), even if all measurements were performed very close to one another in time and space.

## 5. Discussion

### 5.1 Possible errors related to the methodology

Defining a correct algorithm for modeling $SWE$ data is very important for evaluating the water resources deriving from snow melt. The approach applied for deriving $SWE_{SR}$ is highly sensitive to the value used for the new snow density, which can vary substantially depending on both atmospheric and surface conditions. In this way, the error in individual snowfall events could be significant. Moreover, the technique depends on determining snowfall events, which are estimated from changes in snow depth, and the subsequent calculation and accumulation of $SWE_{SR}$ from those events. Therefore, missed events due to gaps in snow depth data could invalidate the calculation of peak $SWE_{SR}$. For these reasons, we focused our analyses on understanding how an incorrect assessment of $\rho_{new\,snow}$ or a gap in snow depth data may affect the $SWE$ estimation.

First, we evaluated the $\rho_{new\,snow}$ estimate (applying Eq. 3, equal to 149 kg m$^{-3}$ considering the 2005-2015 dataset), by means of the leave-one-out cross-validation technique (LOOCV, a particular case of leave-p-out cross-validation with p = 1), to ensure independence between the data we use to estimate $\rho_{new\,snow}$ and the data we use to assess the corresponding estimation error. In this kind of cross-validation, the number of "folds" (repetitions of the cross-validation process) equals the number of observations in the dataset. Specifically, we applied Eq. 3 once for each snow pit ($j$), using all the other snow pits in the calculation ($LOOCV\ \rho_{new\,snow}$) and using the selected snow pit as a single-item test ($\rho_{new\,snow}$ from snow pit $j$). In this way, we avoid dependence between the calibration and validation datasets in assessing the new snow density. The results are shown in Table 2. Analysis shows that the standard deviation of the differences between the $LOOCV\ \rho_{new\,snow}$ values and the corresponding single-item test values ($\rho_{new\,snow}$ from snow pit $j$) is 18 kg m$^{-3}$. The error of the average value of $\rho_{new\,snow}$ can therefore be estimated dividing this standard deviation by the square root of the number of the considered snow pits. It turns out to be 6 kg m$^{-3}$. The new and the old estimates (149 and 140 kg m$^{-3}$, respectively) therefore do not have a statistically significant difference. The individual snow accumulation periods instead have naturally a higher error and the single snow pit estimates for $\rho_{new\,snow}$ range from 128 to 178 kg m$^{-3}$. In addition, we attempted to extend this analysis considering each single snow layer ($h_{ij}$) instead of each snow pit $j$. In particular, we tried to associate with each snow pit layer the corresponding new snow

measured by the sonic ranger (Citterio et al., 2007). However, this approach turned out to be too subjective to contribute accurate information about the $\rho_{new\,snow}$ value we found.

Moreover, we investigated the $SWE$ sensitivity to changes in $\rho_{new\,snow}$. In particular, we calculated $SWE_{SR}$ using different values of new snow density ranging from 100 to 200 kg m$^{-3}$ at 25 kg m$^{-3}$ intervals (Fig. 6). An increase/decrease of the density by 25 kg m$^{-3}$ causes a mean variation in $SWE_{SR}$ of ±106 mm w.e. for each hydrological year (corresponding to about 17% of the mean total cumulative $SWE$ considering all hydrological years), ranging from ±43 mm w.e. to ±144 mm w.e. A reliable estimation of $\rho_{new\,snow}$ is therefore a key issue.

In addition to an accurate definition of new snow density, an uninterrupted dataset of snow depth is also necessary in order to derive correct $SWE_{SR}$ values. This can also be deducted observing the large deviations between the $SWE$ values (independent of the chosen snow density) found by the SR50 and the snow pit measurements in the years 2010, 2011, 2012 and 2013. It is therefore necessary to put in place all the available information to reduce the occurrence of data gaps to a minimum. The introduction of the second sonic ranger (Sommer USH8) at the end of the 2013-2014 snow season was an attempt to limit the impact of this problem. This second sonic ranger, however, was still in the process of testing in the final years of the period investigated in this paper. We are confident that in the years to come it can help reduce the problem of missing data. Indeed, daily variations in snow depth measured by one sensor could be used to fill a data gap from the other one. Multiple sensors for fail-safe data collection are indeed highly recommended. In addition, the four wooden stakes installed at the corners of the snow pillow at the beginning of the 2014-2015 snow season were another idea for collecting more data. Unfortunately, they were broken almost immediately after the beginning of the snow accumulation period. They can offer another way to deal with the problem of missing data, provided we figure out how to avoid breakage during the winter season. Probably the choice of a more robust and white material (such as insulated white steel) could overcome this issue.

It is also important to stress that potential errors in individual snowfall events could affect peak $SWE_{SR}$ estimation. A large snowfall event with a considerable deviation from the mean new snow density will result in significant errors (e.g. a heavy wet snowfall). These events are rather rare at the Forni site: only 3 days in the 11-year period covered by the data recorded more than 40 cm of new snow (the number of days decreases to 1 if the threshold increases to 50 cm). Therefore, even if the proposed technique can be susceptible to these errors, high precipitation amounts are infrequent, reducing the likelihood of this happening at the Forni site. Without knowing the true density of the new snow during these big events, it is difficult to understand their impact on the $SWE$ estimate. However, assuming that the new snow density could be increased from 149 kg m$^{-3}$ to 200 kg m$^{-3}$, the difference in $SWE$ for a large event (e.g. 30 cm) is 15 mm w.e. (45 mm w.e. with 149 kg m$^{-3}$ and 60 mm w.e. with 200 kg m$^{-3}$).

Our new snow data could be affected by settling, sublimation, snow transported by wind, and rainfall. As far as settling is concerned, $\Delta h_{snow-pit-j}$ from Eq. 2 would indeed be higher if $\Delta h_{tj}$ values were calculated considering an interval shorter than 24 hours. However, this would not be possible because on the one hand, the sonic ranger data's margin of error is too high to consider hourly resolution, and on the other hand, new snow is defined by the WMO within the context of a 24-hour period. Settling processes can also concern the snow pack under the new snow layer. This process can affect our daily differences especially when the snowfall lasts for several days. In this case, the measured daily positive snow depth differences could be less than the real depth of the new snow, with the consequence of overestimating new snow density. However, the obtained mean new snow density is not much higher than the general values found in the literature. In addition, comparison with the snow pillow dataset seems to support our methodology. On the other hand, if many days pass between one snowfall

and the following one, the settlement of the snow pack under the new snow layer is less likely to affect the measured
differences in snow depth and this seems to be the case of the Forni Glacier site, since snow days account for only 9% of the
snow season days. Regarding the transport by wind, the effect that is potentially most relevant is new snow that is recorded
by the sonic ranger but then blows away in the following days. It is therefore considered in $\Delta h_{snow-pit-j}$ but not in
$SWE_{snow-pit-j}$, thus causing an underestimation of $\rho_{new\ snow}$ (see Eq. 3). The snow transported to the measuring site can
also influence $\rho_{new\ snow}$, even if in this case the effect is less important, as it is measured both by the sonic ranger and by the
snow pit. Here, the problem may be an overestimation of $\rho_{new\ snow}$ as snow transported by wind usually has a higher density
than new snow. We considered the problem of the effect of wind on snow cover when we selected the station site on the
glacier. Even though sites not affected by wind transport simply do not exist, we are confident that the site we selected has a
position that can reasonably minimize this issue. Moreover, sublimation processes would have an effect similar to those
produced by new snow that is recorded by the sonic ranger but then blown away in the following days. In any case, the value
we found for the site average new snow density (i.e. 149 kg m$^{-3}$) does not seem to suggest an underestimated value.
Finally, another possible source of error in estimating new snow density and in deriving the daily *SWE* is represented by
rainfall events. In fact, one of the effects is an enhanced snow melt and then a decrease in snow depth, as rain water has a
higher temperature than the snow. Therefore, especially at the beginning of the snow accumulation season, we could detect a
snowfall (analyzing snow depth data) but whenever it was followed by a rainfall, the new fallen snow could partially or
completely melt, thus remaining undetected when measured at the end of the accumulation season using snow pit techniques.
This is another potential error that, besides the ones previously considered, could lead to underestimation of the $\rho_{new\ snow}$
value, even if, as already mentioned, the value of 149 kg m$^{-3}$ does not seem to suggest this. On the other hand, rain can also
increase the *SWE* measured using snow pit techniques without giving a corresponding signal in the sonic ranger measurements
of snow depth whenever limited amounts of rain fall over cold snow. In any case, rain events are extremely rare during the
snow accumulation period, so the errors associated with rain are minimal.
### 5.2 Possible errors related to the instrumentation
With regard to the instrumentation, we found some issues related to the derived snow data. Focusing on the beginning of the
snow accumulation period, it appears that neither system of measurement (i.e. sonic ranger and snow pillow) was able to
detect the first snowfall events correctly. With the sonic ranger, the surface roughness of the glacier ice makes it impossible
to distinguish a few centimeters of freshly fallen snow. In fact, the surface heterogeneity (i.e. bare ice, ponds of different size
and depth, presence of dust and fine or coarse debris that can be scattered over the surface or aggregated) translates into a
differential ablation, due to different values of albedo and heat transfer. These conditions cause differences in surface elevation
of up to tens of centimeters and affect the angular distribution of reflected ultrasound. At 3 m of height, the diameter of the
measuring field is 1.17 m for the SR50. For these reasons, the sonic ranger generally records inconsistent distances between
ice surface and sensor, generally much smaller than the values of the previous and subsequent readings. This issue does not
occur with thick snow cover, as the snow roughness is much less than that of ice.
Regarding the snow pillow methodology, analyzing the 2014-2015 and 2015-2016 data, it seems to work correctly only with
a snow cover thicker than 50 cm (Fig. 4). In fact, with null or very low snow depth, *SWE* values are incorrectly recorded. The
results from the snow pillow are difficult to explain as this sensor has been in use for only two winter seasons and we are still
in the process of testing it. Analyzing data from the years to come will strengthen our interpretation. However, we have
searched for a possible explanation of this problem and the error could be due to the configuration of the snow pillow. More-
over, some of the under-measurement or over-measurement errors can commonly be attributed to differences in the amount
of snow settlement over the snow pillow, compared with that over the surrounding ground, or to bridging over the snow pillow
with cold conditions during development of the snow cover (Beaumont, 1965). In addition, another major source of *SWE*
snow pillow errors is generally due to measuring problems of this device, which is sensitive to the thermal conditions of the
sensor, the ground and the snow (Johnson et al., 2015). In fact, according to Johnson and Schaefer (2002) and Johnson (2004)
snow pillow under-measurement and over-measurement errors can be related to the amount of heat conduction from the
ground into the overlying snow cover, the temperature at the ground/snow interface and the insulating effect of the overlying
snow. This particular situation can not be recognized at the Forni Glacier, as the surface consists of ice and not of soil.
Therefore, in our particular case the initial error could be due to the configuration of the snow pillow.
In order to assess the correct outset of the snow accumulation period and overcome the instrument issues, albedo represents a
useful tool, as freshly fallen snow and ice are characterized by very different values (e.g. Azzoni et al., 2016). In fact, whenever
a snowfall event occurs, albedo immediately rises from about 0.2 to 0.9 (typical values of ice and freshly fallen snow, respec-
tively, Senese et al., 2012a). This is also confirmed by the automated camera's hourly pictures. During the hydrological year
2014-2015, the first snowfall was detected on 22 October 2014 by analyzing albedo data, and it is verified by pictures taken
by the automated camera. Before this date, the sonic ranger did not record a null snow depth, mainly due to the ice roughness;
therefore, we had to correct the dataset accordingly.
Concerning the *SWE* as determined by the snow weighing tube, this device is pushed vertically into the snow to fill the tube.
The tube is then withdrawn from the snow and weighed. Knowing the length of tube filled with snow, the cross-sectional area
of the tube and the weight of the snow allows a determination of both the *SWE* and the snow density (Johnson et al., 2015).
The measurements carried out around the AWSs on 20 February 2015 showed a great spatial variability in *SWE* (Fig. 5): the
standard deviation is 54 mm w.e., corresponding to 12% of the mean value from snow weighing tube measurements. This
could explain the differences found analyzing data acquired using the snow pillow techniques, measured by the snow pit, and
derived by the sonic ranger. However, the *SWE* variability highlighted by the snow weighing tube surveys can be also due to
oversampling by this device (Work et al., 1965). Numerous studies have been conducted to verify snow tube accuracy in
determining *SWE*. The most recent studies by Sturm et al. (2010) and Dixon and Boon (2012) found that snow tubes could
under- or over-measure *SWE* from -9% to +11%. Even if we allow for ±10% margin of error in our snow tube measurements,
the high *SWE* variability is confirmed.
Finally, the last approach for measuring *SWE* is represented by the snow pit. This method (like the snow tube) has the down-
side that it is labor intensive and it requires expert personnel. Moreover, as discussed in Senese et al. (2014), it is very im-
portant to select a correct date for making the snow pit surveys in order to assess the total snow accumulation amount. Gen-
erally, 1 April is the date considered the most indicative of the peak cumulative *SWE* in high mountain environments of the
midlatitudes, but this day is not always the best one. In fact, Senese et al. (2014) found that using a fixed date for measuring
the peak cumulative *SWE* is not the most suitable solution. In particular, they suggest that a correct temperature threshold can
help to determine the most appropriate time window of analysis, indicating the starting time of snow melting processes and
then the end of the accumulation period. From the Forni Glacier, the application of the +0.5°C daily temperature threshold
allows for a consistent quantification of snow ablation while, instead, for detecting the beginning of the snow melting pro-
cesses, a suitable threshold has proven to be at least −4.6°C. A possible solution to this problem could be to repeat the snow
pit surveys over the same period to verify the variability of microscale conditions. This can be useful especially in those
remote areas where no snowfall information is available. However, this approach involves too much time and resources and
is not always feasible.
Even if the generally used sensors (such as the heated tipping bucket rain gauge, the heated weighing gauge, or the disdrom-
eter) provide more accurate measurements, in remote areas like a glacier, it is very difficult to install and maintain them. One
of the limitations concerns the power to be supplied to instruments, which can only consist in solar panels and lead-gel bat-
teries. In fact, at the Forni site we had to choose only unheated low-power sensors. The snow pillow turned out to be logisti-
cally unsuitable, as it required frequent maintenance. Especially with bare ice or few centimeters of snow cover, the differen-
tial ablation causes instability of the snow pillow, mainly due to its size. Therefore, the first test on this sensor seems to
indicate that it did not turn out to be appropriate for a glacier surface. We will, however, try to get better results from it in the
coming years. The snow pit can represent a useful approach but it requires expert personnel for carrying out the measurements,
and the usefulness of the data so-obtained depends on the date for excavating the snow pits. The automated camera provided
hourly photos, but for assessing a correct snow depth at least two graduated rods have to be installed close to the automated
camera. However, over a glacier surface, glacier dynamics and snow flux can compromise the stability of the rods: in fact, at
the AWS Forni SPICE we found them broken after a short while. Finally, the SR50 sonic ranger features the unique problem
of the definition of the start of the accumulation period, but this can be overcome using albedo data.

## 6. Conclusions

For the SPICE project, snow measurements at the Forni Glacier (Italian Alps) have been implemented by means of several
automatic and manual approaches since 2014. This has allowed an accurate comparison and evaluation of the pros and cons
of using the snow pillow, sonic ranger, snow pit, or snow weighing tube, and of estimating $SWE$ from snow depth data. We
found that the mean new snow density changes based on the considered period was: 140 kg m$^{-3}$ in 2005-2009 (Senese et al.,
2014) and 149 kg m$^{-3}$ in 2005-2015. The difference is, however, not statistically significant. We first evaluated the new snow
density estimation by means of LOOCV and we found an error of 6 kg m$^{-3}$. Then, we benchmarked the derived $SWE_{SR}$ data
against the information from the snow pillow (data which was not used as input in our density estimation), finding a RMSE
of 45 mm w.e. (corresponding to 8% of the maximum $SWE$ measured by means of the snow pillow). These analyses permitted
a correct definition of the reliability of our method in deriving $SWE$ from snow depth data. Moreover, in order to define the
effects and impacts of an incorrect $\rho_{new\,snow}$ value in the derived $SWE$ amount, we found that a change in density of ±25 kg m$^{-3}$
causes a mean variation of 17% of the mean total cumulative $SWE$, considering all hydrological years. Finally, once $\rho_{new\,snow}$
is known, the sonic ranger can be considered a suitable device on a glacier, or in a remote area in general, for recording
snowfall events and for measuring snow depth values in order to derive $SWE$ values. In fact, the methodology we have pre-
sented here can be interesting for other sites as it allows estimating total $SWE$ using a relatively inexpensive, low power, low
maintenance, and reliable instrument such as the sonic ranger, and it is a good solution for estimating $SWE$ at remote locations
such as glacier or high alpine regions. In addition, our methodology ensured that the mean new snowfall density can be reliably
estimated.
Although conventional precipitation sensors, such as the heated tipping bucket rain gauges, heated weighing gauges or dis-
drometers, can perhaps provide more accurate estimates of precipitation and $SWE$ than the ones installed at the Forni Glacier,
they are less than ideal for use in high alpine and glacier sites. The problem is that in remote areas like a glacier at a high
alpine site, it is very difficult to install and maintain them. The main constrictions concern i) the power supply to the instru-
ments, which consists in solar panels and lead-gel batteries, and ii) the glacier dynamics, snow flux and differential snow/ice
ablation that can compromise the stability of the instrument structure. Therefore, a sonic ranger could represent a useful
approach for estimating *SWE*, since it does not require expert personnel, nor does it depend on the date of the survey (as do
such manual techniques as snow pits and snow weighing tubes); it is not subject to glacier dynamics, snow flux or differential
ablation (as are graduated rods installed close to an automated camera and snow pillows), and it does not required a lot of
power (unlike heated tipping bucket rain gauges). The average new snow density must, however, be known either by means
of snow pit measurements or by the availability of information from similar sites in the same geographic area.

**Acknowledgements**
The AWS1 Forni was developed under the umbrella of the SHARE (Stations at High Altitude for Research on the Environ-
ment) program, managed by the Ev-K2-CNR Association from 2002 to 2014; it was part of the former CEOP network (Co-
ordinated Energy and Water Cycle Observation Project) promoted by the WCRP (World Climate Research Programme)
within the framework of the online GEWEX project (Global Energy and Water Cycle Experiment); it was inserted in the
SPICE (Solid Precipitation Intercomparison Experiment) project managed and promoted by the WMO (World Meteorological
Organization), and in the CryoNet project (core network of Global Cryosphere Watch promoted by the WMO), and it was
applied in the ESSEM COST Action ES1404 (a European network for a harmonised monitoring of snow for the benefit of
climate change scenarios, hydrology and numerical weather prediction).
This research has been carried out under the umbrella of a research project funded by Sanpellegrino Levissima Spa, and young
researchers involved in the study were supported by the DARAS (Department of regional affairs, autonomies and sport) of
the Presidency of the Council of Ministers of the Italian government through the GlacioVAR project (PI G. Diolaiuti). More-
over, the Stelvio Park - ERSAF kindly supported data analyses and has been hosting the AWS1 Forni and the AWS SPICE
at the surface of the Forni Glacier thus making possible the launch of glacier micro-meteorology in Italy.
The authors are acknowledge the Special Issue Editor Mareile Wolff for her help in improving the first draft of this paper and
the two reviewers for their useful comments and suggestions. The authors are also grateful to Carol Rathman for checking
and improving the English language of this manuscript.

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

Table 1: Instrumentation at the Forni Glacier with instrument name, measured parameter, manufacturer, and starting date.

| Instrument name | Parameter | Manufacturer | Date |
|---|---|---|---|
| Babuc ABC | Data logger | LSI LASTEM | Sept. 2005 |
| CR200 | Data logger | Campbell | May 2014 |
| CR1000 | Data logger | Campbell | May 2014 |
| Sonic ranger SR50 | Snow depth | Campbell | Sept. 2005 |
| Sonic ranger USH8 | Snow depth | Sommer | May 2014 |
| Snow pillow | SWE | Park Mechanical Inc. | May 2014 |
| Thermo-hygrometer | Air temperature and humidity | LSI LASTEM | Sept. 2005 |
| Barometer | Atmospheric pressure | LSI LASTEM | Sept. 2005 |
| Net Radiometer CNR1 | Short and long wave radiation fluxes | Kipp & Zonen | Sept. 2005 |
| Pluviometer unheated | Liquid precipitation | LSI LASTEM | Sept. 2005 |
| Anemometer 05103V | Wind speed and direction | Young | Sept. 2005 |


Table 2: The leave-one-out cross-validation (LOOCV). For each survey, we reported the *SWE* values measured by means of
the snow pit (*SWE$_{snow\text{-}pit}$*), the values of the new snow density applying the Eq. 3 ($\rho_{new\ snow}$ from snow pit *j*), and the new snow
density obtained applying the LOOCV method (*LOOCV $\rho_{new\ snow}$*).

| Date of survey | $SWE_{snow\text{-}pit}$ (mm w.e.) | $\rho_{new\ snow}$ from snow pit *j* (kg m$^{-3}$) | LOOCV $\rho_{new\ snow}$ (kg m$^{-3}$) |
|---|---|---|---|
| 24/01/06 | 337 | 147 | 150 |
| 02/03/06 | 430 | 128 | 153 |
| 30/03/06 | 619 | 147 | 150 |
| 07/05/08 | 690 | 135 | 152 |
| 21/02/09 | 650 | 143 | 151 |
| 27/03/10 | 640 | 156 | 149 |
| 25/04/11 | 770 | 178 | 147 |
| 20/02/15 | 555 | 159 | 149 |
| **MEAN** | | **149** | **150** |


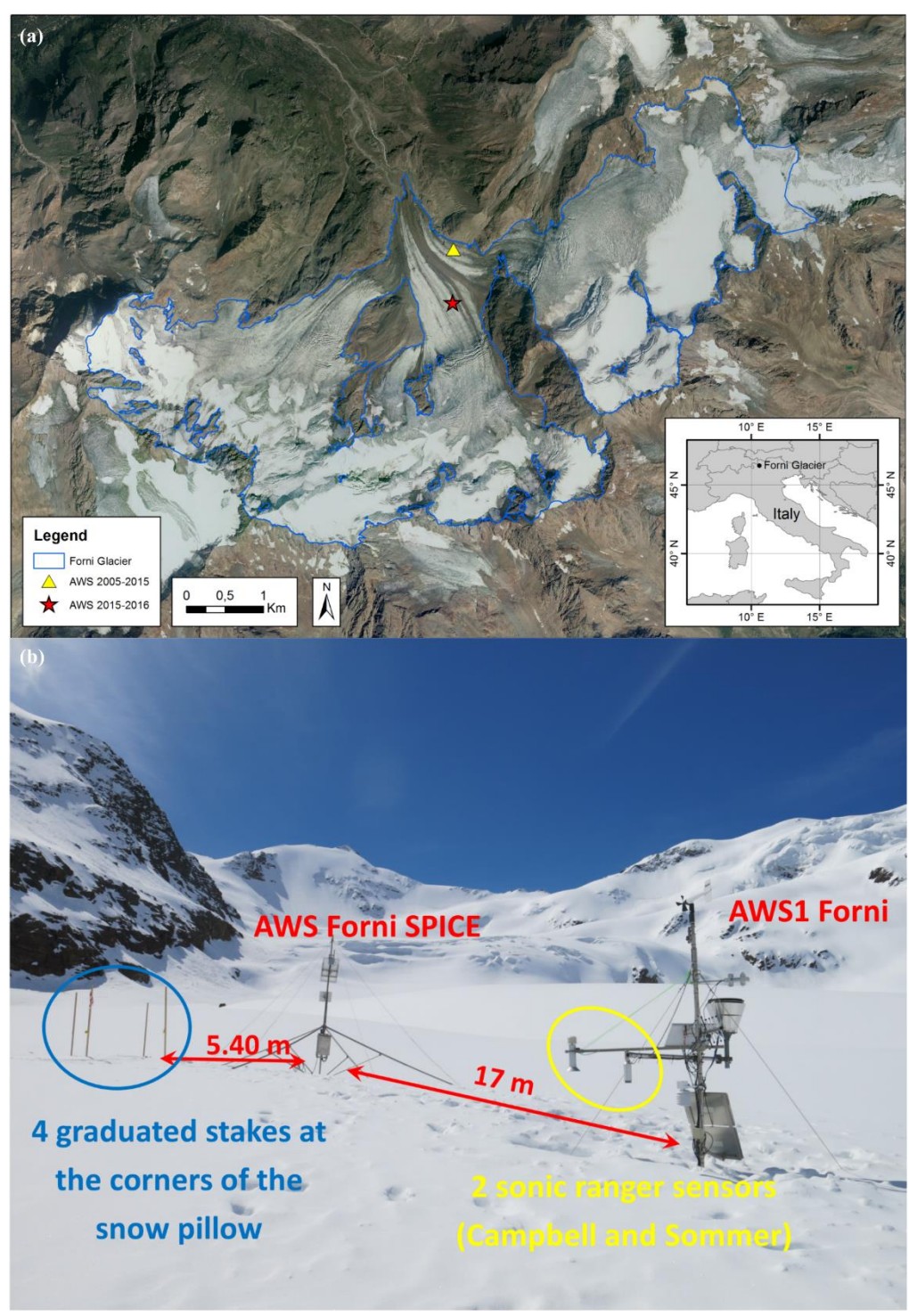


Figure 1: (a) The study site. The yellow triangle indicates the location of the AWS1 Forni and the Forni AWS SPICE
until November 2015. The red star refers to the actual location after securing the stations. (b) AWS1 Forni (on the
right) and AWS Forni SPICE (on the left) photographed from the North-East on 6th May 2014 (immediately after the
installation of the AWS Forni SPICE). The distances between the stations are shown.

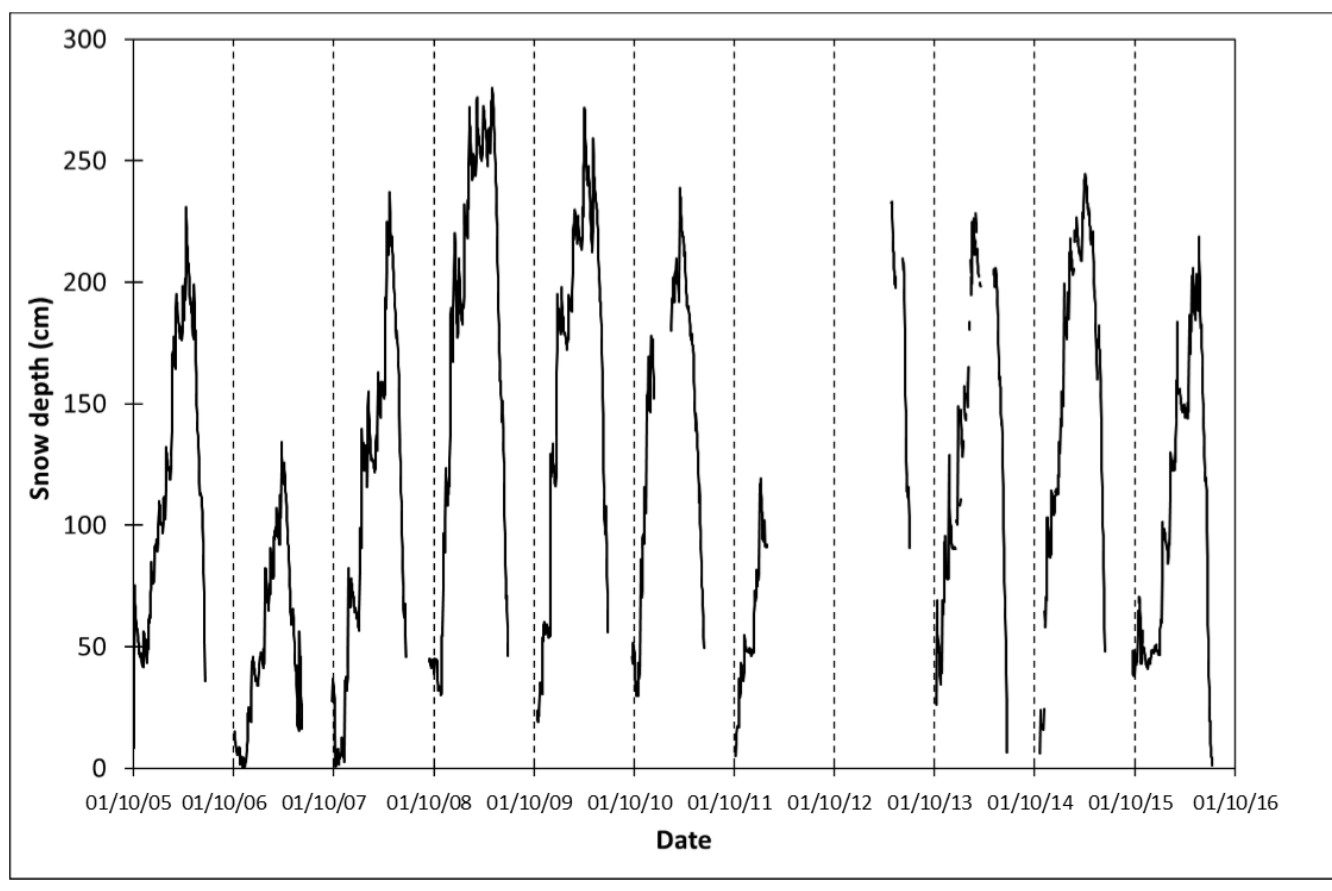

Figure 2: Daily snow depth measured by the Campbell SR-50 sonic ranger at the AWS1 Forni from 1st October 2005 to 30th September 2016. The dates shown are dd/mm/yy.


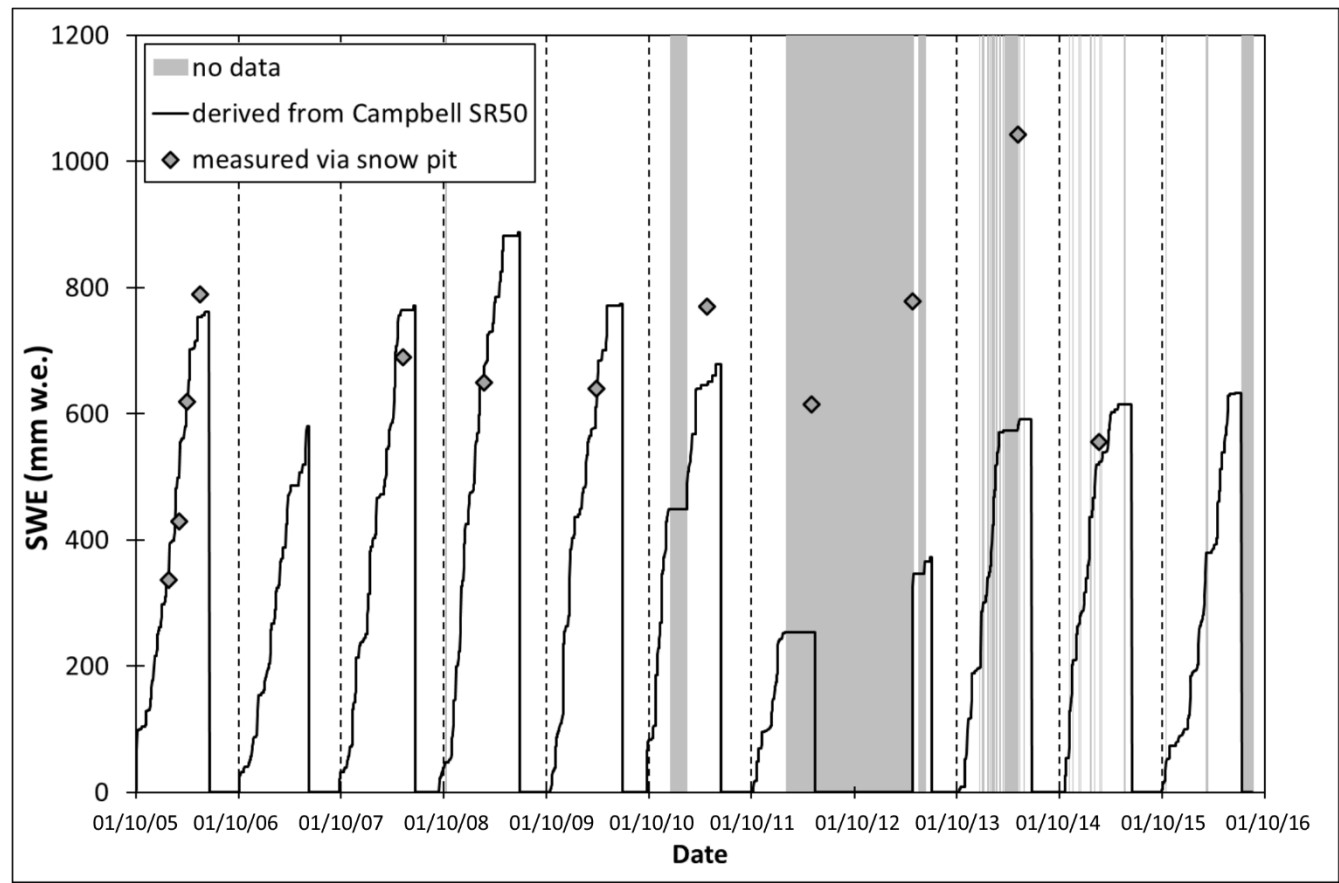

**Figure 3: Daily *SWE* data derived from snow depth by the Campbell SR50 (using the new snow density of 149 kg m$^{-3}$)**


**and measured by snow pits from 1$^{st}$ October 2005 to 30$^{th}$ September 2016. The periods without data are shown in light**


**grey. The dates shown are dd/mm/yy.**



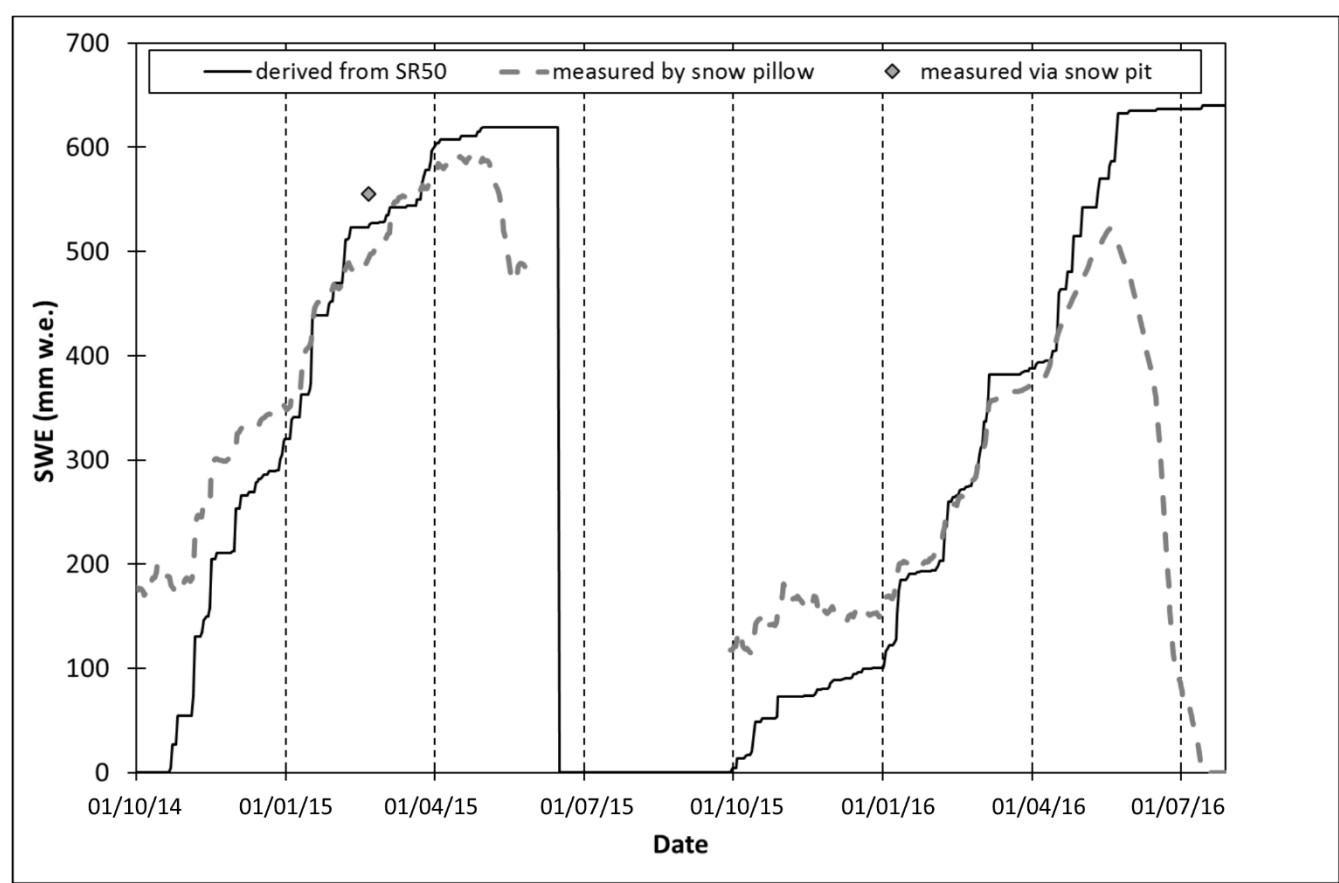


**Figure 4: Daily SWE data derived from snow depth measured by Campbell SR50 (using the new snow density of 149**
**kg m$^{-3}$) and measured by snow pits and snow pillow from October 2014 to July 2016. The dates shown are dd/mm/yy.**

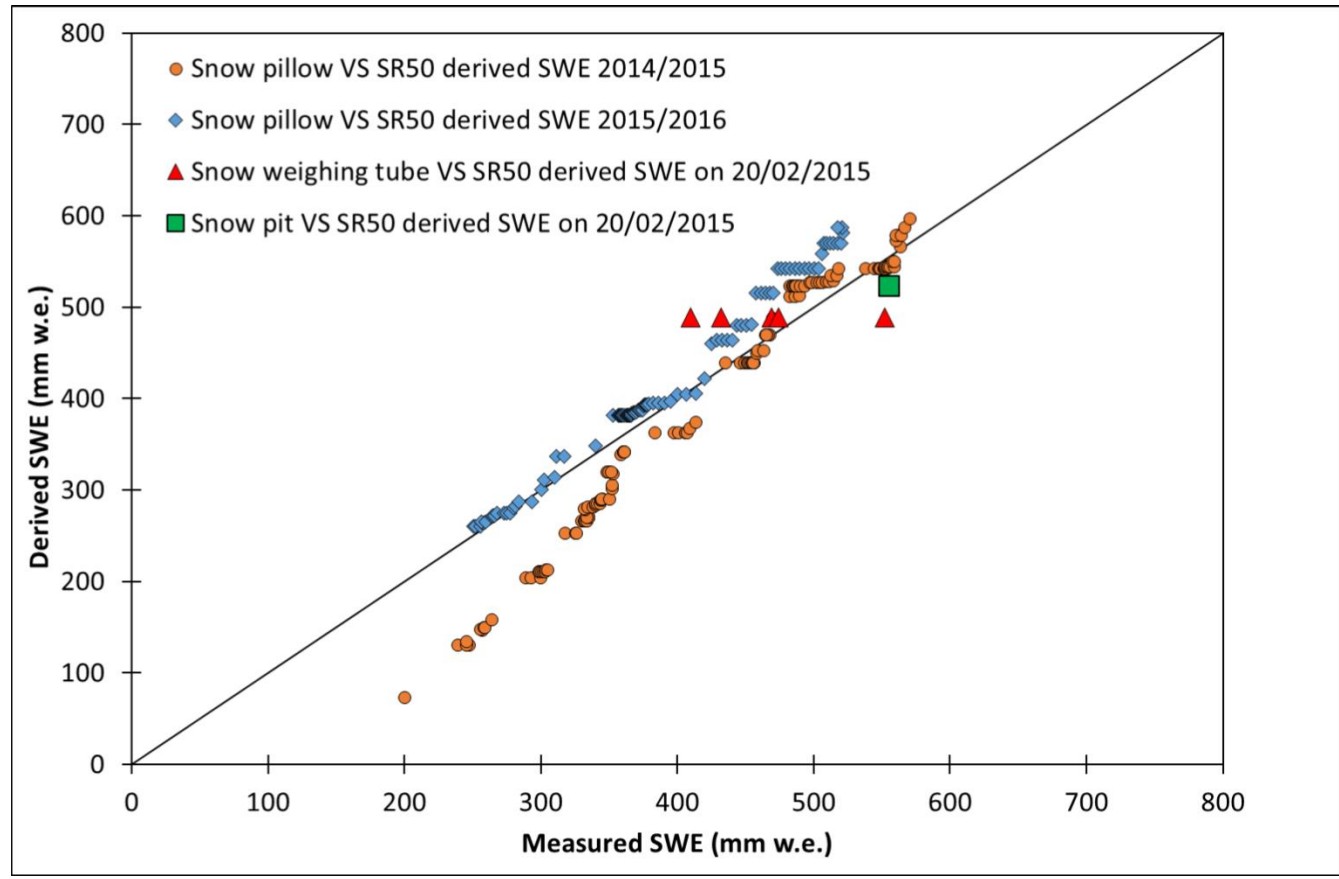


**Figure 5: Scatter plots showing SWE measured by snow pillow and snow pit and derived applying Eq. (4) to data**
**acquired by Campbell SR50 (using the new snow density of 149 kg m$^{-3}$). Two accumulation periods of measurements**
**are shown from November 2014 to March 2015 and from February 2016 to May 2016. Every dot represents a daily**
**value.**

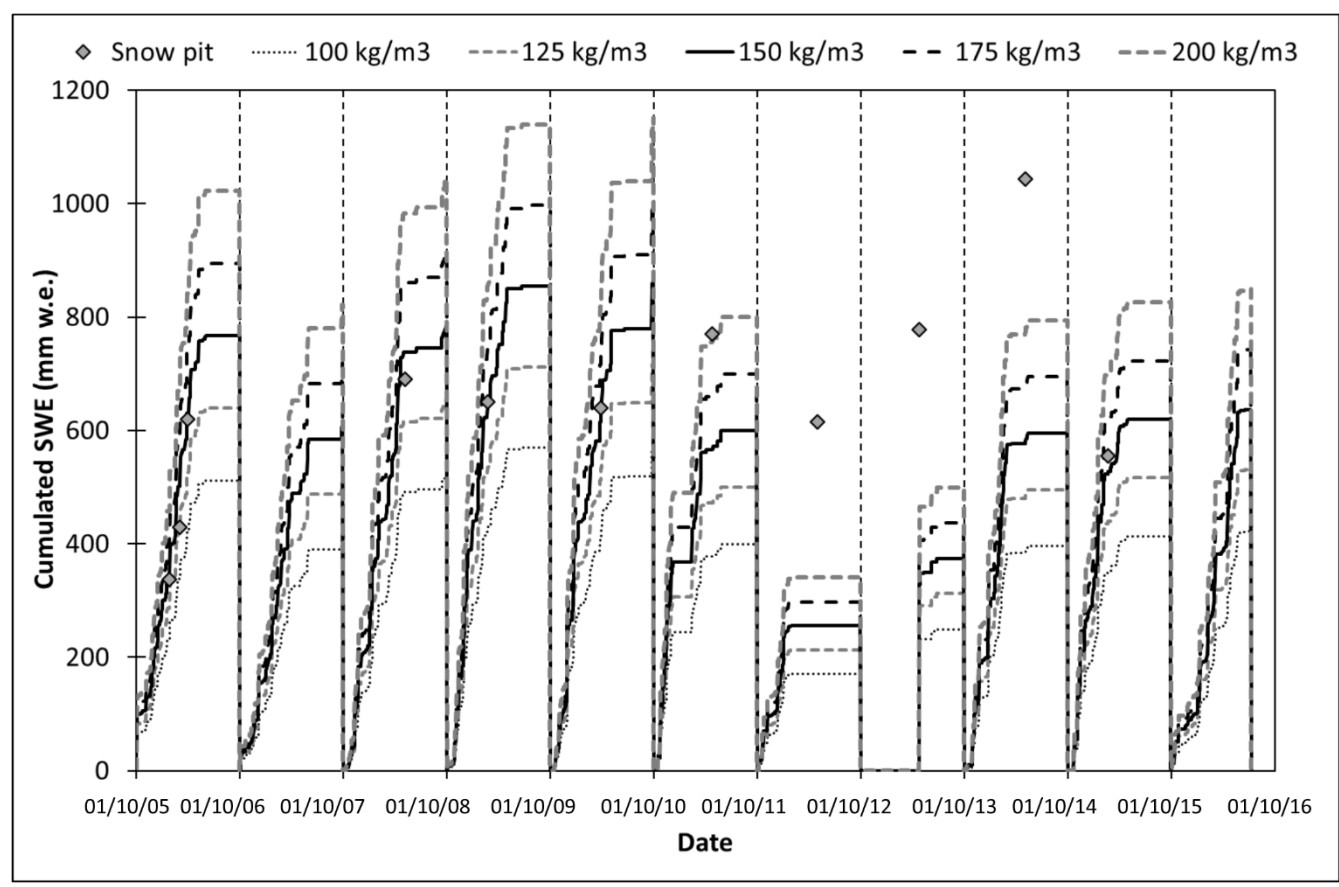


**Figure 6: Comparison among daily *SWE* values derived from snow depth data acquired by SR50 sonic ranger (apply-**

**ing different values of new snow density) and *SWE* values measured by snow pits from 2005 to 2016. The dates shown**
**are dd/mm/yy.**