# Peer review of "Estimating the snow water equivalent on glacierized high elevation areas (Forni Glacier, Italy)"

_The Cryosphere, 2017_

## Referee Comment (RC1) · C. Smith (Referee) · 22 Aug 2017

**Review of Senese et al (2017) "Snow data intercomparison on remote and glacierized high elevation areas (Forni Glacier, Italy), August 18, 2017**

The paper by Senese et al. presents a methodology for calculating **accumulating** snow water equivalent (SWE) at a remote site located on a high elevation glacier in northern Italy. The method uses an average new snowfall density estimated for the site and calculates accumulating SWE on an event by event basis from snow depths measured by the automated snow depth sensor. The accumulated SWE is then compared to estimates made from periodic snow pits (usually only once per season near peak accumulation) and snow pillow measurements. Since the focus of the paper (as written and as per my suggestions below) is on SWE, I would suggest revising the title to "Estimating snow water equivalent on remote and glacierized high elevation areas (Forni Glacier, Italy)".

The technique is interesting in that it provides an estimate, albeit a potentially crude estimate, for total SWE using a relatively inexpensive, low power, low maintenance, and reliable (in my opinion) instrument and is presented as an alternative for estimating SWE at remote locations such as glacier monitoring and research in high alpine regions. The technique itself is filled with potential issues (and the authors point some of these out) and should only be used to produce an approximation of total SWE.

 I have a concern about how the average new snowfall density is estimated. The way I understand it from the methodology, density is backwards calculated from snow depth and pit derived SWE using the same data as shown in the intercomparison (Fig 4) so the intercomparison between snow pit SWE and depth derived SWE is not independent. An independent estimate of average new snow density would have been more appropriate. An alternative would be to calculate the average new snow density from the first couple of seasons and then apply this to the remaining seasons. The lack of independence should be stated clearly and the intercomparison should focus on the snow pillow data as independent (mostly) validation.

As I mentioned, and as the authors point out, the technique has some potential issues: 1) it is highly sensitive to the estimation of average new snow density, which can vary substantially depending on both atmospheric and surface conditions, so the error in individual events could be quite large and therefore the value is only in the estimation of peak SWE and 2) the technique depends on determining event based changes in snow depth and the subsequent calculation and accumulation of SWE from those events, so missed events due to gaps in snow depth data invalidates the calculation of peak SWE.

My biggest concern with this paper is that the issues are not adequately addressed in the discussion. For example, the discussion of issues related to the use of a mean snow density is only a few lines long and should be expanded substantially. Discuss the potential errors in individual events and how this impacts your peak estimation. A large event (big increase in snow depth) that has a large deviation from the mean density will result in larger errors (e.g. a heavy wet snowfall). What is the potential for this to occur at this site? Add a discussion about missing data as this is the greatest threat to failure of the

technique. Can you do gap filling with photographed snow stakes? Would you recommend redundant sensors? More specific comments are listed below.

Note that the units for SWE should be reported in mm water equivalent (w.e.) or kg m$^{-2}$ and not m w.e. . Snow depths should also be reported in cm and not m. It would also be useful if you used an abbreviation for "sonic ranger-derived SWE" such as SWE$_{SR}$ and use this throughout the paper.

**Abstract**

Page 1, Line 15: "…on the Forni Glacier **in Italy**."

> Lines 19-20: This sentence misses the mark. From what I have read, you are not really assessing the mean value of new snow density…this was done elsewhere. I think you miss explicitly stating the aim of the analysis and the value of this paper. You should state here that you are using mean new snowfall density and automated depth measurements to estimate the SWE of new snowfall and accumulating to estimate peak SWE and evaluating against other methods.

> Line 21: "rather good" is a vague and subjective description of the estimation. Avoid this and/or quantify the estimation.

**Introduction**

Page 2, Line 37: "…**often** only snowfall measure**ment**s are available…"

> Line 38: "assess" should be "calculate" and "depending" should be "depends". Fresh snow density also depends on surface conditions, correct?

> Line 50: CryoNet is more of a network attached to the GCW initiative, rather than a "project"

> Line 71: "detail" not "details"

Page 3, Lines 89-93: For item ii, I'm not sure that you are "defining the reliability of…" because you don't really have a solid reference (more on that later) to be able to do that. I would rather you said that you were "assessing the capability of…". You also use the term "obtained SWE" here…now I assume that you are referring to the derived SWE from the depth measurements so you need to be more specific here.

**Data and Methods**

Page 3, Line 100: You refer to "These sensors" but you should rather say "These measurements were made at the two stations…" since not all of the measurements were made with sensors.

> Line 106: Was the T/RH sensor shielded? I assume yes.

Page 4, Line 110: You should cite Beaumont (1965) here when introducing the snow pillow methods. Also some grammar issues with the first sentence on this page.

Line 116: I think "constrictions" should be "challenges".

Line 121: "represented" should be "provided" or "supplied". "deep" should be "thorough".

Line 122: "working" should perhaps be "operation".

Line 123: "…due to ice flow, etc.) **is required** before installation…"

Lines 125-128: Not sure if this paragraph adds anything to the paper

Line 130: I think that I know what you mean by "adjust" but you should clarify this.

Line 132: What are ring faults?

Line 146-147: "…from snow depth acquired by sonic rangers **and estimated new snowfall density**." Clarify what you mean by the last sentence "In particular…". This is a crucial piece of the methodology and you should describe this better. What time did you use for start/end? Did you do any noise filtering? How do you account for redistribution, settling, and sublimation during the day? (should also be discussed in the Discussion section).

Page 5, Lines 152-153:  What do you mean by "unique date"?

Lines 150-153: This is the methodology that isn't clearly described (see note above).

**Results**

Beside expressing the errors or biases in absolute units, it would be helpful if the relative bias was stated. Same can be said for subsequent sections.

Lines 159-161: Can you comment here or in the discussion on any potential impacts on the analysis due to the site move.

Lines 166-169: The intercomparison between the SR50 and the USH8 is interesting but largely irrelevant for this paper. I would omit this. What you could mention is that the two instruments had a correlation of ?? and that from 2015 onwards, redundancy in the snow depth measurement could mean better data for the SWE estimate.

Line 174: What is "a very good agreement"?

Line 180: "elaborate" should be "accumulate"

Figure 4: it would be very useful if the missing SR50 data could be indicated (e.g. different colour line, etc). This would certainly help with the interpretation of the graph. One might argue that all of the estimated SWE should be set to missing after large gaps in the snow depth data since it is an accumulated result.

Page 6, Line 188: "…thus suggesting a correct working of the sensor." This is more than a sensor, but rather a technique or process. This sentence is awkward and should be reworded.

Line 192: typo "derided".

Line 194: Change "raises" to "increases".

Lines 196-197: You mention the snow course data using the snow tube and suggest a "large spatial variability". Perhaps you could report the average and standard deviation of this data and in the discussion, relate this variability to the differences you see between the SR50 SWE estimation and the snow pillow and snow pit.

Lines 200-201: Oversampling of the tube is a potential error and should be included in the discussion and not in the results. You need to estimate this error and put it into context with the differences between the methods.

**Discussion**

The start of the Discussion paragraph needs a couple of sentences of introduction to "frame" the problem and set up the following paragraphs.

Lines 205-208: The whole success of technique hinges on the approximation of the average new snowfall density but yet this only gets a few sentences in the discussion. This needs to be expanded. When would you expect errors to be greatest? At this site, can you potentially get large snowfalls that have much greater (or lower) densities than the average, which could then bias your accumulated SWE? Any suggestions on how to better estimate new snowfall SWE, perhaps at sites where you don't have snow pits to back-calculate the density.

Line 205: "Once our procedure was verified, we performed…"

Line 209: What is a "general good agreement". Quantify this.

Line 214: "not constant" should be "inconsistent"

Page 7, Lines 223-231: You reference some issues with the snow pillow and then state that your snow pillow only seems to be working with depths greater that 50cm. Can you relate any of the referenced potential issues to the errors that you are seeing at the site? I'm not convinced that your issues are related to those referenced so this discussion needs to be stronger.

Lines 232-246: This paragraph is more of a justification for using this technique and is more appropriate for the introduction than the discussion. The sentences (Lines 246-250) about the snow pillow is relevant, however. The subsequent discussion about the snow pit should have its own paragraph.

Line 248-249: You should refer to Fig 5 and discuss this error in context with what you are seeing at the site. This should be combined with the snow pillow discussion. Also, is there a reference for this statement about minimum snow pillow measurements?

Line 252: What do you mean by "whole glacier accumulation amount"?

Line 253: "…the **peak** cumulative SWE…" Same applies for the next sentence.

Line 257: Typo "closed" should be "close"

Lines 259-261: The two sentences beginning with "Finally…" seem out of place or awkward. Please re-organize these.

Page 8, Lines 261-263: The discussion about the SR50/USH8 is largely unsubstantiated and should be omitted.

I would like to see me recommendations in the discussion about how to make this technique better and how to adapt it for other remote sites. Perhaps make some recommendations on further testing, possibly to take advantage of other data collected during SPICE.

**Conclusions**

Line 267:  Was the acronym SPICE not defined earlier in the paper? If it was, it doesn't need to be defined here.

Line 269: The sentence beginning with "This has allowed…" needs some commas to read correctly.

Line 270: "…SWE values using a fresh snow density…"

Lines 270-273:  I have some problems with the statement "The results achieved…". You only tested two sensors for measuring snow depth so I wouldn't go as far as saying that the SR50 is the "most suitable device"…perhaps it is but you don't have enough information to substantiate this. You present a technique for estimating peak SWE on a glacier and you compare this technique against another technique (e.g. the snow pillow) but you don't really have a true reference so I'm not sure you can say that it is "most suitable". Rather, I would reference your relative error and despite the issues, suggest that the technique is "suitable" for estimating SWE.

Line 274: What are the relative variations as a result of the density change?

Instead of ending the conclusions with a comment about the error, perhaps end with a comment on the applicability to other remote sites.

---

## Referee Comment (RC2) · Anonymous Referee #2 · 13 Sep 2017

**Review of**

**Snow data intercomparison on remote and glacierized high elevation areas (Forni Glacier, Italy)**

The paper describes a method how to estimate SWE from automatic snow depth measurements at a high alpine site on a glacier during the accumulation phase. The method uses a fixed density for each daily positive snow depth difference to calculate the increasing SWE until its maximum. The method seems interesting and may have some potential.

My main criticism concerns the following important facts, which are missing:

- the exact determination of the fixed density
- the calculation of the daily positive snow depth differences
- the limitations and uncertainty of the method
- transferability to other sites and climates

The fixed density of 140 kg/m3 has already been used in earlier papers (2012, 2104) from you. Indeed the comparison between the estimated SWE from snow depth measurements and SWE from snow pits (the current Fig. 4) has already been published in The Cryosphere in 2014 (Fig. 2 in https://www.the-cryosphere.net/8/1921/2014/tc-8-1921-2014.pdf). However, there the estimated SWE data seem to be different from the one shown in the current paper? Can you explain?

Here a list of other important points, which need to be addressed before I can recommend accepting the paper for publication:

- Your fixed density is not a fresh snow density, since you totally neglect settling and are not able to determine small snow falls due to the measurement uncertainty of the snow depth sensor. The found density of 140 kg/m3 can therefore not be compared with published fresh snow densities found in literature and is relatively large since it has to compensate the missing snow fall amounts mentioned above.
- The impact of rain events at the beginning and end of the snow season has not been discussed so far.
- Since your focus is the determination SWE from snow depth your title needs to be changed to something like: Estimation of SWE from automatic snow depth measurements during accumulation on a high alpine glacier.
- The content of the abstract is odd. The abstract needs to be rewritten.
- The possible impact of the dislocation of the station needs to be discussed.
- You need to present some numbers about the uncertainty of the involved sensors, manual measurements and the uncertainty of the presented results. That means the papers definitely needs more quantitative information about the performance of your method.
- Information about the measurement frequency and aggregation of the data shown in the figures need to included.
- A comparison with data from other sites is needed in order to be able to judge the usefulness of the method.
- The English language of the paper is often odd and needs to be improved in revised version.

A list of minor comments and correction has directly been written in PDF of the paper below.

[revised manuscript text omitted]

---

## Author Comment (AC1) · 25 Oct 2017

**Dear Referee**

Thanks for helping us in improving the manuscript.

We will consider all your comments in the revised version of the manuscript. The title, as you suggest and in agreement also with the second Referee, will be: Estimating snow water equivalent on remote and glacierized high elevation areas (Forni Glacier, Italy).

We will therefore modify the "Method" section in order to better explain the approach applied for estimating the site-average-new-snow density ( $\rho_{new snow}$ ) and the sonic-ranger-depth-derived *SWE*. In the previous version of the manuscript, we performed the analyses using the mean new snow density (140 kg m-3) that was obtained by Senese et al. (2012) considering the 2005-2009 dataset, and then we discussed to what degree this value is able to describe the data of the following years. In the revised version of the manuscript, we will start with updating the site average new snow density estimation exploiting all the available dataset and we will perform all the subsequent analyses using this value.

The text concerning this issue in "Method" section will therefore be:

In addition to the measures performed by means of the AWSs, since winter 2005-2006, personnel from the Centro Nivo-Meteorologico (namely CNM Bormio-ARPA Lombardia) of the Lombardy Regional Agency for the Environment have been carrying out periodic snow pits (performed according to the AINEVA protocol, see also Senese et al., 2014) in order to estimate snow depth and *SWE*. In particular, for each snow pit *j*, the thickness ( $h_{ij}$ ) and the density ( $\rho_{ij}$ ) of each snow layer (*i*) are measured for determining its snow water equivalent then the total *SWE*snow-pit-j of the whole snow cover (*n* layers) is obtained:

$$SWE_{snow-pit-j} = \sum_{i=1}^{n} h_{ij} \cdot \frac{\rho_{ij}}{\rho_{water}}$$

(1)

where  $\rho_{water}$  is density of water. As stated in a previous study (Senese et al., 2014), the date when the snow pit is dug is very important for not underestimating the actual accumulation. For this reason, we considered only the snow pits carried out before the beginning of snow ablation. In fact, whenever ablation occurs, successive *SWE* values derived from snow pits show a decreasing trend (i.e. they are affected by mass losses).

The snow pit *SWE* data were then used, together with the corresponding total new snow derived from sonic ranger measures, to estimate the site average  $\rho_{new snow}$ , in order to update the value of 140 kg m-3 that was found in a previous research on data of the same site covering the 2005-2009 period (Senese et al., 2012a). We need updating  $\rho_{new snow}$  as it is the key variable for estimating *SWE* from sonic ranger new snow data. Specifically, for each snow pit *j*, the corresponding total new snow was first determined by:

$$\Delta h_{snow-pit-j} = \sum_{t=1}^{m} (\Delta h_{tj})$$
(2)

where *m* is the total number of days with snowfall in the period corresponding to snow pit *j* and  $\Delta h_{ij}$  corresponds to the depth of new snow of day *t*. In particular, we considered the hourly snow depth values recorded by the sonic ranger in a day and we calculated the difference between the last and the first reading. Whenever this difference is positive (at least 1 cm), it corresponds to a new snowfall. All data are subject to a strict control to avoid under- or over-measurements, to remove outliers and non-sense values, and to filter possible noises.  $\sum_{t=1}^{m} (\Delta h_{tj})$  is therefore the total new snow measured by the Campbell SR50 from the beginning of the accumulation period to the date of the snow pit survey. The average site  $\rho_{new snow}$  was then determined as:

$$\rho_{new\ snow} = \frac{\sum_{j=1}^{k} SWE_{snow-pit-j}}{\sum_{j=1}^{k} (\Delta h_{snow-pit-j})}$$
(3)

where *j* identifies a given snow pit and the corresponding total new snow and the sum extends over all *k* available snow pits. Instead of a mere average of  $\rho_{new snow}$  values obtained from individual snow pit surveys, this relation gives more weight to snow pits with a higher *SWE*snow-pit amount.

The *SWE* of each day (t) was then estimated by:

$$SWE_{SR-t} = \begin{cases} \Delta h_t \frac{\rho_{new\,snow}}{\rho_{water}} & \text{if } \Delta h_t \ge 1 \ cm \\ 0 & \text{if } \Delta h_t < 1 \ cm \end{cases}$$
(4)

We will also apply the leave-one-out cross-validation (LOOCV, a particular case of leave-p-out cross-validation with p = 1) to ensure independence between the data we use to estimate  $\rho_{new snow}$  and the data we use to assess the corresponding estimation error. Specifically, we will apply Eq. (3) (see answer above) once for each snow pit (*j*), using all other snow pits in the relation ( $\rho_{new snow}$  LOOCV) and using the selected snow pit as a single-item test ( $\rho_{new snow}$  from snow pit *j*). In this way, we will avoid dependence between calibration and validation dataset in assessing the new snow density.

The results will give evidence that the standard deviation of the differences between the  $\rho_{new snow}$  LOOCV values and the corresponding single-item test values ( $\rho_{new snow}$  from snow pit *j*) is 18 kg m-3. The error of the average value of  $\rho_{new snow}$  will then be estimated dividing this standard deviation for the square root of the number of the considered snow pits. We will show that it is 6 kg m-3. We will finally show that the new and the old estimates of  $\rho_{new snow}$  (149 and 140 kg m-3, respectively) do not have a statistical significant difference.

We will also discuss this issue focusing on the snow pit layers (see Eq. (1) – answer above).

Finally, before uploading the reviewed manuscript, as suggested by the second Referee, the standard of English spelling and grammar will be improved by a professional, mother-tongue consultant.

Point-by-point answers to your comments follow.

Discuss the potential errors in individual events and how this impacts your peak estimation. A large event (big increase in snow depth) that has a large deviation from the mean density will result in larger errors (e.g. a heavy wet snowfall). What is the potential for this to occur at this site? Add a discussion about missing data as this is the greatest threat to failure of the technique. Can you do gap filling with photographed snow stakes? Would you recommend redundant sensors? More specific comments are listed below.

We will address these issues in the "Discussion" section of the new version of the manuscript.

As far as large events are concerned, they are rather rare at the studied site: only 3 days in the 11-year period covered by the data recorded more than 40 cm of new snow (the number of days decreases to 1 if the threshold increases to 50 cm). More in detail, we have the following distribution of new snow: 382 days with values between 1 and 10 cm, 95 days with values between 10 and 20 cm, 33 days with values between 20 and 30 cm, 11 days with values between 30 and 40 cm. Beside investigating the distribution of new snow values, we checked also if the days in the different new snow intervals have significantly different average temperatures. We did not find any signal.

We agree that missing data is a relevant issue. The introduction of the second sonic ranger (Sommer USH8) at the end of the snow season 2013-2014 was an attempt to reduce the impact of this problem. The second sonic ranger, however, was still in its process of testing in the last years of the period investigated within this paper (we e.g. changed the sensor model). We are confident that in the next years it can help reducing the problem of missing data. Redundant sensors are indeed highly recommended.

The four stakes installed at the corners of the snow pillow at the beginning of the 2014-2015 snow season were another idea to have more data. Unfortunately, they were broken almost immediately after the beginning of the snow accumulation period. They can be another way to face the problem of missing data, provided we will find out how to avoid they will break during the winter season.

Finally, we will add in the revised version of the paper some comments about the relevance of the single snow pit layers to the total snow pit SWE.

Note that the units for SWE should be reported in mm water equivalent (w.e.) or kg m-2 and not m w.e. . Snow depths should also be reported in cm and not m. It would also be useful if you used an abbreviation for "sonic ranger-derived SWE" such as  $SWE_{SR}$  and use this throughout the paper.

We will modify all the manuscript accordingly.

**Abstract**

Page 1, Line 15: "...on the Forni Glacier in Italy."

We will modify the sentence accordingly.

Lines 19-20: This sentence misses the mark. From what I have read, you are not really assessing the mean value of new snow density...this was done elsewhere. I think you miss explicitly stating the aim of the analysis and the value of this paper. You should state here that you are using mean new snowfall density and automated depth measurements to estimate the SWE of new snowfall and accumulating to estimate peak SWE and evaluating against other methods.

We will modify the sentence accordingly.

*Line 21: "rather good" is a vague and subjective description of the estimation. Avoid this and/or quantify the estimation.*

We have calculate the RMSE between SWE derived from SR50 sonic ranger and measured by means of snow pillow and we have found a RMSE of 45 mm w.e. We will add this information in the abstract accordingly.

**Introduction**

Page 2, Line 37: "...often only snowfall measurements are available..."

We will modify the sentence accordingly.

*Line 38: "assess" should be "calculate" and "depending" should be "depends". Fresh snow density also depends on surface conditions, correct?*

We will modify the sentences accordingly. We will add "surface conditions" in addition to "atmospheric conditions".

Line 50: CryoNet is more of a network attached to the GCW initiative, rather than a "project"

We will modify the sentence accordingly.

Line 71: "detail" not "details"

We will modify the sentence accordingly.

Page 3, Lines 89-93: For item ii, I'm not sure that you are "defining the reliability of..." because you don't really have a solid reference (more on that later) to be able to do that. I would rather you said that you were "assessing the capability of...". You also use the term "obtained SWE" here...now I assume that you are referring to the derived SWE from the depth measurements so you need to be more specific here.

We will modify the sentence accordingly: "ii) assess the capability to obtain SWE values from the depth measurements".

**Data and Methods**

Page 3, Line 100: You refer to "These sensors" but you should rather say "These measurements were made at the two stations..." since not all of the measurements were made with sensors.

We will modify the sentence accordingly.

Line 106: Was the T/RH sensor shielded? I assume yes.

The thermo-hygrometer is with shield. We will modify the sentence accordingly adding "shielded".

Page 4, Line 110: You should cite Beaumont (1965) here when introducing the snow pillow methods. Also some grammar issues with the first sentence on this page.

We will modify the sentence accordingly citing Beaumont (1965) and checking the grammar issues.

Line 116: I think "constrictions" should be "challenges". We will modify the sentence accordingly.

*Line 121: "represented" should be "provided" or "supplied". "deep" should be "thorough".* We will modify the sentences accordingly.

Line 122: "working" should perhaps be "operation". We will modify the sentence accordingly.

Line 123: "...due to ice flow, etc.) is required before installation..." We will modify the sentence accordingly.

*Lines 125-128: Not sure if this paragraph adds anything to the paper* We will delete this paragraph accordingly.

*Line 130: I think that I know what you mean by "adjust" but you should clarify this.* We will modify the sentence accordingly in order to better explain what we mean.

**Line 132: What are ring faults?**

The ring faults develop as a series of circular or semicircular fractures with stepwise subsidence, caused by englacial or subglacial meltwater creating voids at the ice-bedrock interface and eventually the collapse of cavity roofs.

We will add the definition in the new version of the manuscript.

Line 146-147: "...from snow depth acquired by sonic rangers and estimated new snowfall density." Clarify what you mean by the last sentence "In particular...". This is a crucial piece of the methodology and you should describe this better. What time did you use for start/end? Did you do any noise filtering? How do you account for redistribution, settling, and sublimation during the day? (should also be discussed in the Discussion section).

In order to estimate the "daily positive differences in depth ( $\Delta h$ )", we considered the hourly snow depth recorded by the sonic ranger in a day and we calculated the difference between the last and the first reading. Whenever this difference is positive, it corresponds to a possible new snowfall:

$$\Delta h = \begin{cases} reading at h23 - reading at h00 (if positive) \\ 0 (if negative) \end{cases}$$

This issue will be better explained in the revised version of the manuscript (see answers to the main comments).

As far as settling is concerned,  $\Delta h_{snow-pit-j}$  from eq. 2 (see answers to the main comments) would indeed be higher if  $\Delta h_{tj}$  were calculated considering a shorter interval than 24 hours. However, on one side this would not be possible because the sonic ranger data have too high error to consider hourly resolution, and on the other side, new snow is defined by WMO considering a 24-hour period. In our opinion, therefore, we do not need considering settling as new snow defined by WMO does already include the settling that occurs in the period used to measure new snow.

Obviously, our sonic ranger data may be affected by snow transported by wind. The effect that is potentially more relevant is new snow that is recorded by the sonic ranger and then blow away in the following days. It is therefore considered in  $\Delta h_{snow-pit-j}$  and not in  $SWE_{snow-pit-j}$  which causes an underestimation of  $\rho_{new snow}$  (see eq. 3 – answers to the main comments). Also snow transported to the measuring site can influence  $\rho_{new snow}$  even if in this case the effect is lower as it measured both by the sonic ranger and by the snow pit. In this case, the problem may be an overestimation of  $\rho_{new snow}$  as snow transported by wind has usually higher density than new snow. We considered the problem of the effect of wind on snow cover when we selected the station site over the glacier. Even though sites that are not affected by wind transport do not exist, we are confident that the site we selected has a position that can reasonably lower this issue.

Sublimation would have an effect that is similar to those produced by new snow that is recorded by the sonic ranger and then blow away in the following days. We did not yet investigate it. We underline however that the value we find for the site average new snow (i.e. 149 kg m-3) does not seem to suggest an underestimated value.

As regards the quality-check activities, we submitted the dataset to a strict control to avoid under- or overmeasurements, to remove outliers and non-sense values, to filter possible noises. We will modify the paragraph in the "Method" section accordingly in order to better explain our approach (see answer to the main comments).

**Page 5, Lines 152-153: What do you mean by "unique date"?**

We will modify the sentence accordingly in order to better clarify the approach (see answer to the main comments).

**Lines 150-153: This is the methodology that isn't clearly described (see note above).**

In the new version of the manuscript, we will modify this part of the "Method" section (following also the suggestions of the second Referee) in order to better clarify our procedure in estimating the density of the new snow ( $\rho_{new snow}$ ) used for deriving SWE from snow depth data (see answers to the main comments).

**Results**

Beside expressing the errors or biases in absolute units, it would be helpful if the relative bias was stated. Same can be said for subsequent sections.

We will modify the previously showed values accordingly adding the error.

*Lines 159-161: Can you comment here or in the discussion on any potential impacts on the analysis due to the site move.*

The dislocation of the AWSs could influence snow conditions, as there could be a different snow accumulation due to a different radiation input and diverse wind regimes. However, as the distance between the two sites is about 500 m, the difference in elevation is only 44 m and the aspect is very similar, we do not expect a noticeable impact of the site change on snow depth. However, we will add some discussions about the site move in the revised manuscript following also the suggestions of the second Referee.

Lines 166-169: The intercomparison between the SR50 and the USH8 is interesting but largely irrelevant for this paper. I would omit this. What you could mention is that the two instruments had a correlation of ?? and that from 2015 onwards, redundancy in the snow depth measurement could mean better data for the SWE estimate.

We will omit the intercomparison and the relative figure accordingly. We will move this part into the "Method" section, specifying the correlation between the datasets from Campbell and Sommer sensors.

Line 174: What is "a very good agreement"?

We will add the root mean square error, that is equal to 58 mm w.e.

*Line 180: "elaborate" should be "accumulate"*

We will modify the sentence accordingly.

Figure 4: it would be very useful if the missing SR50 data could be indicated (e.g. different colour line, etc). This would certainly help with the interpretation of the graph. One might argue that all of the estimated SWE should be set to missing after large gaps in the snow depth data since it is an accumulated result.

We will modify the figure showing the period without snow depth data. The new version of the figure is:

Page 6, Line 188: "...thus suggesting a correct working of the sensor." This is more than a sensor, but rather a technique or process. This sentence is awkward and should be reworded.

We will modify the sentence accordingly.

Line 192: typo "derided".

We will modify in "SWESR" in according to your previous comment.

Line 194: Change "raises" to "increases".

We will modify the sentence accordingly.

Lines 196-197: You mention the snow course data using the snow tube and suggest a "large spatial variability". Perhaps you could report the average and standard deviation of this data and in the discussion, relate this variability to the differences you see between the SR50 SWE estimation and the snow pillow and snow pit.

We will modify accordingly adding the mean value of 165 cm and the standard deviation of 29 cm. In addition, as reported in the following comment, we will discuss this variability in the "Discussion" section.

*Lines 200-201: Oversampling of the tube is a potential error and should be included in the discussion and not in the results. You need to estimate this error and put it into context with the differences between the methods.*

As stated in Johnson et al. (2015), numerous studies have been conducted to determine snow tube accuracy in determining SWE (Freeman, 1965; Work et al., 1965; Beaumont, 1967; Peterson and Brown, 1975; Goodison, 1978; Farnes et al., 1982), the most recent being by Sturm et al. (2010) and Dixon and Boon (2012). The most recent comparison of snow tubes by Dixon and Boon (2012) examined the Standard Federal, Meteorological Service of Canada, and SnowHydro snow tubes. The earlier studies had found that snow tubes tended to overmeasure SWE from about 4–11%, whereas the recent studies by Sturm et al. (2010) and Dixon and Boon (2012) found that snow tubes could under-measure or over-measure SWE from -9% to 11%. Even if we consider that our snow tube measures can be affected by errors of about  $\pm$  10%, the high SWE variability is confirmed.

This part will be discussed in the "Discussion" section of the revised manuscript.

**Discussion**

The start of the Discussion paragraph needs a couple of sentences of introduction to "frame" the problem and set up the following paragraphs.

We will add a paragraph accordingly in order to better introduce the "Discussion" section. The initial part of the "Discussion" section will therefore state that defining a correct algorithm for modelling *SWE* data is very important to evaluate the water resource deriving from snow melt. We will also highlight that the approach we suggest to derive *SWE*SR is highly sensitive to the value used for the new snow density, which can vary substantially depending on both atmospheric and surface conditions. In this way, the error of SWE from individual snowfall events could be quite large. Moreover, the technique depends on determining snowfall events, that are estimated from changes in snow depth, and the subsequent calculation and accumulation of *SWE*SR from those events. Therefore, missed events due to gaps in snow depth data could invalidate the calculation of peak *SWE*SR.

This first part of the "Discussion" section will explain why we analyzed how an incorrect assessment of  $\rho_{new snow}$  or a gap in snow depth data may affect the estimation of the *SWE*.

Lines 205-208: The whole success of technique hinges on the approximation of the average new snowfall density but yet this only gets a few sentences in the discussion. This needs to be expanded. When would you expect errors to be greatest? At this site, can you potentially get large snowfalls that have much greater (or lower) densities than the average, which could then bias your accumulated SWE? Any suggestions on how to better estimate new snowfall SWE, perhaps at sites where you don't have snow pits to back-calculate the density.

We will modify this paragraph accordingly. In particular, for validating our procedure in estimating the new snow density, we will apply the leave-one-out cross-validation (LOOCV, a particular case of leave-p-out cross-validation with p = 1) in order to assess both the error of the estimation of the average site  $\rho_{new snow}$  and the error we perform if we estimate  $\rho_{new snow}$  of each single snow pit by means of our approach.

In addition, we will investigate the *SWE* sensitivity to changes in  $\rho_{new snow}$ : we will calculate *SWE*SR using different values of new snow density ranging from 100 to 200 kg m-3 with a step of 25 kg m-3.

We will also add to the "Discussion" section some new information on the occurrence of outliers in our data set (the highest new snow value is slightly higher than 40 cm) and on the distribution of new snow. We will then underline that we do not have a significant link between new snow and corresponding daily average temperature that could e.g. indicate that higher new snow values are more frequent at the beginning and at the end of the snow season. Finally, we will present a first analysis of the snow pit.

Line 205: "Once our procedure was verified, we performed..."

We will modify this paragraph accordingly to the previous comment.

**Line 209: What is a "general good agreement". Quantify this.**

We will modify the sentence accordingly from "Beside to a general good agreement between the measures performed with the different sensors, there are also some problems." to "As regards the instrumentations, we could assure the correct working of all sensors. Nevertheless, we found some issues related to the derived snow information".

**Line 214: "not constant" should be "inconsistent"**

We will modify the sentence accordingly.

Page 7, Lines 223-231: You reference some issues with the snow pillow and then state that your snow pillow only seems to be working with depths greater that 50cm. Can you relate any of the referenced potential issues to the errors that you are seeing at the site? I'm not convinced that your issues are related to those referenced so this discussion needs to be stronger.

The results from the snow pillow are difficult to explain as this sensor has been working for only two winter seasons and then we are still in the process of testing. Analyzing data of the next years will allow a more robust interpretation. However, we have searched for a possible explanation of this over-weighing accordingly and perhaps this error may be due to the configuration of the snow pillow.

*Lines 232-246: This paragraph is more of a justification for using this technique and is more appropriate for the introduction than the discussion. The sentences (Lines 246-250) about the snow pillow is relevant, however. The subsequent discussion about the snow pit should have its own paragraph.*

We will modify this paragraph accordingly, moving part of it into the "Introduction" section. In addition, we will discuss about the snow pit measurements in a separate paragraph.

Line 248-249: You should refer to Fig 5 and discuss this error in context with what you are seeing at the site. This should be combined with the snow pillow discussion. Also, is there a reference for this statement about minimum snow pillow measurements?

We will modify this section as described in the previous comment.

**Line 252: What do you mean by "whole glacier accumulation amount"?**

We will modify the sentence accordingly from "whole glacier accumulation amount" to "total snow accumulation amount".

*Line 253: "...the peak cumulative SWE..." Same applies for the next sentence.* We will modify the sentences accordingly.

**Line 257: Typo "closed" should be "close"**

We will modify the sentence accordingly.

*Lines 259-261: The two sentences beginning with "Finally..." seem out of place or awkward. Please re-organize these.*

We will modify the sentences accordingly from "Finally, with data acquired by the SR50 sonic ranger a correct curve of SWE was derived. The unique issue is represented by the definition of the beginning of the accumulation period, but this can be overcome using albedo data." to "In spite of all these issues, the SR50 sonic ranger features the unique limit represented by the definition of the beginning of the accumulation period, but this can be overcome using albedo data." to "In spite of all these issues, the SR50 sonic ranger features the unique limit represented by the definition of the beginning of the accumulation period, but this can be overcome using albedo data."

Page 8, Lines 261-263: The discussion about the SR50/USH8 is largely unsubstantiated and should be omitted. We will delete the discussion regarding the USH8 sonic ranger accordingly.

I would like to see me recommendations in the discussion about how to make this technique better and how to adapt it for other remote sites. Perhaps make some recommendations on further testing, possibly to take advantage of other data collected during SPICE.

We will modify the "Discussion" section accordingly, following all your previous suggestions and comments.

**Conclusions**

*Line 267: Was the acronym SPICE not defi

---

## Author Comment (AC2) · 25 Oct 2017

Dear Referee

Thanks for helping us in improving the manuscript.

We will consider all your comments in the revised version of the manuscript. The title, as you suggest and in agreement also with the second Referee, will be: Estimating snow water equivalent on remote and glacierized high elevation areas (Forni Glacier, Italy).

We will therefore modify the "Method" section in order to better explain the approach applied for estimating the site-average-new-snow density ($\rho_{new\ snow}$) and the sonic-ranger-depth-derived *SWE*. In the previous version of the manuscript, we performed the analyses using the mean new snow density (140 kg m$^{-3}$) that was obtained by Senese et al. (2012) considering the 2005-2009 dataset, and then we discussed to what degree this value is able to describe the data of the following years. In the revised version of the manuscript, we will start with updating the site average new snow density estimation exploiting all the available dataset and we will perform all the subsequent analyses using this value.

The text concerning this issue in "Method" section will therefore be:

In addition to the measures performed by means of the AWSs, since winter 2005-2006, personnel from the Centro Nivo-Meteorologico (namely CNM Bormio-ARPA Lombardia) of the Lombardy Regional Agency for the Environment have been carrying out periodic snow pits (performed according to the AINEVA protocol, see also Senese et al., 2014) in order to estimate snow depth and *SWE*. In particular, for each snow pit *j*, the thickness ($h_{ij}$) and the density ($\rho_{ij}$) of each snow layer (*i*) are measured for determining its snow water equivalent then the total *SWE$_{snow-pit-j}$* of the whole snow cover (*n* layers) is obtained:

$$SWE_{snow-pit-j} = \sum_{i=1}^{n} h_{ij} \cdot \frac{\rho_{ij}}{\rho_{water}} \tag{1}$$

where $\rho_{water}$ is density of water. As stated in a previous study (Senese et al., 2014), the date when the snow pit is dug is very important for not underestimating the actual accumulation. For this reason, we considered only the snow pits carried out before the beginning of snow ablation. In fact, whenever ablation occurs, successive *SWE* values derived from snow pits show a decreasing trend (i.e. they are affected by mass losses).

The snow pit *SWE* data were then used, together with the corresponding total new snow derived from sonic ranger measures, to estimate the site average $\rho_{new\ snow}$, in order to update the value of 140 kg m$^{-3}$ that was found in a previous research on data of the same site covering the 2005-2009 period (Senese et al., 2012a). We need updating $\rho_{new\ snow}$ as it is the key variable for estimating *SWE* from sonic ranger new snow data. Specifically, for each snow pit *j*, the corresponding total new snow was first determined by:

$$\Delta h_{snow-pit-j} = \sum_{t=1}^{m} (\Delta h_{tj}) \tag{2}$$

where *m* is the total number of days with snowfall in the period corresponding to snow pit *j* and $\Delta h_{tj}$ corresponds to the depth of new snow of day *t*. In particular, we considered the hourly snow depth values recorded by the sonic ranger in a day and we calculated the difference between the last and the first reading. Whenever this difference is positive (at least 1 cm), it corresponds to a new snowfall. All data are subject to a strict control to avoid under- or over-measurements, to remove outliers and non-sense values, and to filter possible noises. $\sum_{t=1}^{m} (\Delta h_{tj})$ is therefore the total new snow measured by the Campbell SR50 from the beginning of the accumulation period to the date of the snow pit survey. The average site $\rho_{new\ snow}$ was then determined as:

$$\rho_{new\ snow} = \frac{\sum_{j=1}^{k} SWE_{snow-pit-j}}{\sum_{j=1}^{k} (\Delta h_{snow-pit-j})} \tag{3}$$

where *j* identifies a given snow pit and the corresponding total new snow and the sum extends over all *k* available snow pits. Instead of a mere average of $\rho_{new\ snow}$ values obtained from individual snow pit surveys, this relation gives more weight to snow pits with a higher *SWE$_{snow-pit}$* amount.

The *SWE* of each day (*t*) was then estimated by:

$$SWE_{SR-t} = \begin{cases} \Delta h_t \frac{\rho_{new\ snow}}{\rho_{water}} & if\ \Delta h_t \geq 1\ cm \\ 0 & if\ \Delta h_t < 1\ cm \end{cases} \tag{4}$$

We will also apply the leave-one-out cross-validation (LOOCV, a particular case of leave-p-out cross-validation with p = 1) to ensure independence between the data we use to estimate $\rho_{new\ snow}$ and the data we use to assess the corresponding estimation error. Specifically, we will apply Eq. (3) (see answer above) once for each snow pit (*j*), using all other snow pits in the relation ($\rho_{new\ snow}$ LOOCV) and using the selected snow pit as a single-item test ($\rho_{new\ snow}$ from snow pit *j*). In this way, we will avoid dependence between calibration and validation dataset in assessing the new snow density.

The results will give evidence that the standard deviation of the differences between the $\rho_{new\ snow}$ LOOCV values and the corresponding single-item test values ($\rho_{new\ snow}$ from snow pit $j$) is 18 kg m$^{-3}$. The error of the average value of $\rho_{new\ snow}$ will then be estimated dividing this standard deviation for the square root of the number of the considered snow pits. We will show that it is 6 kg m$^{-3}$. We will finally show that the new and the old estimates of $\rho_{new\ snow}$ (149 and 140 kg m$^{-3}$, respectively) do not have a statistical significant difference.

We will also discuss this issue focusing on the snow pit layers (see Eq. (1) – answer above).

Finally, before uploading the reviewed manuscript, as suggested by the second Referee, the standard of English spelling and grammar will be improved by a professional, mother-tongue consultant.

Point-by-point answers to your comments follow.

*- the exact determination of the fixed density*

In the new version of the manuscript, we will modify this part of the "Method" section (following also the suggestions of the first Referee) in order to better clarify our procedure in estimating the density of the average new snow ($\rho_{new\ snow}$) used for deriving SWE from snow depth data (see answers to the main comments).

*- the calculation of the daily positive snow depth differences*

In order to estimate the "daily positive differences in depth ($\Delta h$)", we considered the hourly snow depth recorded by the sonic ranger in a day and we calculated the difference between the last and the first reading. Whenever this difference is positive, it corresponds to a possible new snowfall:

$$\Delta h = \begin{cases} reading\ at\ h23 - reading\ at\ h00\ (if\ positive) \\ 0\ (if\ negative) \end{cases}$$

This issue will be better explained in the revised version of the manuscript (see answers to the main comments).

*- the limitations and uncertainty of the method*

For validating our procedure in estimating the new snow density, we will apply the leave-one-out cross-validation (LOOCV, a particular case of leave-p-out cross-validation with p = 1) in order to assess both the error of the estimation of the average site $\rho_{new\ snow}$ and the error we perform if we estimate $\rho_{new\ snow}$ of each single snow pit by means of our approach (see answers to the main comments).

In addition, we will investigate the *SWE* sensitivity to changes in $\rho_{new\ snow}$: we will calculate $SWE_{SR}$ using different values of new snow density ranging from 100 to 200 kg m$^{-3}$ with a step of 25 kg m$^{-3}$.

We will also add to the "Discussion" section some new information on the occurrence of outliers in our data set (the highest new snow value is slightly higher than 40 cm) and on the distribution of new snow. We will then underline that we do not have a significant link between new snow and corresponding daily average temperature that could e.g. indicate that higher new snow values are more frequent at the beginning and at the end of the snow season. Finally, we will present a first analysis of the snow pit.

Moreover, we will investigate if potential errors in individual snowfall events could affect peak $SWE_{SR}$ estimation. In fact, a large snowfall event with a large deviation from the mean new snow density will result in larger errors (e.g. a heavy wet snowfall). Large events are rather rare at the studied site: only 3 days in the 11-year period covered by the data recorded more than 40 cm of new snow (the number of days decreases to 1 if the threshold increases to 50 cm). More in detail, we have the following distribution of new snow: 382 days with values

between 1 and 10 cm, 95 days with values between 10 and 20 cm, 33 days with values between 20 and 30 cm, 11 days with values between 30 and 40 cm. Beside investigating the distribution of new snow values, we checked also if the days in the different new snow intervals have significantly different average temperatures. We did not find any signal.

Another possible source of error in estimating new snow density and in deriving daily SWE is represented by rainfall event. In fact, one of the effects is an enhanced snow melt and then a decrease in snow depth, as rain water has a higher temperature than the snow. Therefore, especially at the beginning of the snow accumulation season, we could detect a snowfall (analyzing snow depth data) but, whenever it is followed by a rainfall, the fallen new snow could partially or completely melt and then it could not be measured by snow pit carried out at the end of the accumulation season.

In addition to the validation of new snow density estimation, we will benchmark derived SWE data against the ones measured by the snow pillow (data not used as input in our density estimation). This validation will permit to correctly define the reliability of our method in deriving SWE from snow depth data.

All these data elaborations and analyses will be included in the "Discussion" section.

*- transferability to other sites and climates*

In our opinion, the methodology we presented can be interesting for other sites as it allows estimating total *SWE* using a relatively inexpensive, low power, low maintenance, and reliable instrument such as the sonic ranger and it is a good solution for estimating *SWE* at remote locations such as glacier or high alpine regions. We will highlight this point in the Conclusions of the revised manuscript.

*The fixed density of 140 kg/m3 has already been used in earlier papers (2012, 2104) from you. Indeed the comparison between the estimated SWE from snow depth measurements and SWE from snow pits (the current Fig. 4) has already been published in The Cryosphere in 2014 (Fig. 2 in https://www.the-cryosphere.net/8/1921/2014/tc-8-1921-2014.pdf). However, there the estimated SWE data seem to be different from the one shown in the current paper? Can you explain?*

The values shown in the previous paper are the same but cumulating daily SWE data the yearly end date is different. In this study, we showed the SWE values only during the snow accumulation period, neglecting the ice ablation period. In fact, in this period the fallen snow is completely melted within a few days and always before the beginning of the accumulation period. Instead, in Senese et al. (2014) we cumulated daily SWE values until 30[th] September of each year. Only for this reason, the two figures seem different, but they show the same datasets.
However, in the revised manuscript, we will show SWE data derived using the update new snow density (149 kg m$^{-3}$, instead of 140 kg m$^{-3}$)

*- Your fixed density is not a fresh snow density, since you totally neglect settling and are not able to determine small snow falls due to the measurement uncertainty of the snow depth sensor. The found density of 140 kg/m3 can therefore not be compared with published fresh snow densities found in literature and is relatively large since it has to compensate the missing snowfall amounts mentioned above.*

We will modify our terminology accordingly from "fresh snow density" to "new snow density". In addition, we agree that missing data is a relevant issue. The introduction of the second sonic ranger (Sommer USH8) at the end of the snow season 2013-2014 was an attempt to reduce the impact of this problem. The second sonic

ranger, however, was still in its process of testing in the last years of the period investigated within this paper (we e.g. changed the sensor model). We are confident that in the next years it can help reducing the problem of missing data. Redundant sensors are indeed highly recommended.

The four stakes installed at the corners of the snow pillow at the beginning of the 2014-2015 snow season were another idea to have more data. Unfortunately, they were broken almost immediately after the beginning of the accumulation period. They can be another way to face the problem of missing data, provided we will find out how to avoid they will break during the winter season.

*- The impact of rain events at the beginning and end of the snow season has not been discussed so far.*

We will add some discussion about the impact of rain events. In fact, another possible source of error in estimating new snow density and in deriving daily SWE is represented by rainfall event. One of the effects is an enhanced snow melt and then a decrease in snow depth, as rain water has a higher temperature than the snow. Therefore, especially at the beginning of the snow accumulation season, we could detect a snowfall (analyzing snow depth data) but, whenever it is followed by a rainfall, the fallen new snow could partially or completely melt and then it could not be measured by snow pit carried out at the end of the accumulation season. This is therefore another potential error that, besides to the ones considered in the answers to the first Referee, could potentially give an underestimated value of $\rho_{new\,snow}$. We again highlight however that the value we find (149 kg m$^{-3}$) does not seem to suggest an underestimation.

*- Since your focus is the determination SWE from snow depth your title needs to be changed to something like: Estimation of SWE from automatic snow depth measurements during accumulation on a high alpine glacier.*

Following also the suggestions of the first Referee, we will modify the title accordingly from "Snow data intercomparison on remote and glacierized high elevation areas (Forni Glacier, Italy)" to "Estimating snow water equivalent on remote and glacierized high elevation areas (Forni Glacier, Italy)".

*- The content of the abstract is odd. The abstract needs to be rewritten.*

Following also the suggestions of the first Referee, we will modify the abstract accordingly.

*- The possible impact of the dislocation of the station needs to be discussed.*

The dislocation of the AWSs could influence snow conditions, as there could be a different snow accumulation due to a different radiation input and diverse wind regimes. However, as the distance between the two sites is about 500 m, the difference in elevation is only 44 m and the aspect is very similar, we do not expect a noticeable impact of the site change on snow depth. However, we will add some discussions about the site move in the revised manuscript following also the suggestions of the first Referee.

*- You need to present some numbers about the uncertainty of the involved sensors, manual measurements and the uncertainty of the presented results. That means the papers definitely needs more quantitative information about the performance of your method.*

Following also the suggestions of the first Referee, we will add and discuss the uncertainty of the method for deriving the new snow density and of all the measurements and techniques applied in this study for quantifying *SWE*:
- The average site new snow density is found to be affected by an error of 6 kg m$^{-3}$ (by means of the leave-one-out cross-validation).
- The comparison between SWE derived by snow depth data and SWE measured by means of the snow pillow showed a RMSE of 45 mm w.e.
- Snow tubes could under-measure or over-measure SWE from about -10% to about +10% (as found by Sturm et al., 2010, and Dixon and Boon, 2012).

*- Information about the measurement frequency and aggregation of the data shown in the figures need to included.*

Data points are sampled at 60-second intervals and averaged over a 60-minute time period for SR50 sonic ranger, wind sensor and barometer, over a 30-minute time period for the sensors recording air temperature, relative humidity, solar and infrared radiation, and liquid precipitation, and over a 10-minute time period for USH8 sonic ranger and snow pillow. All data are recorded in a flash memory card, including the basic distribution parameters (minimum, mean, maximum, and standard deviation values). We will add all these information in the revised manuscript accordingly.
In addition, we will add in all the figure captions the aggregation of data.

*- A comparison with data from other sites is needed in order to be able to judge the usefulness of the method.*

Comparing our results with snow depth data from 10 stations spread over the Italian Alps, the snow depth peaks are in agreement with findings over the Italian Alps. In fact, analyzing data in the period 1960–2009 from 10 stations above 1500 m a.s.l. Valt and Cianfarra (2010) reported a mean snow depth of 233 cm (from 199 to 280 cm). At the Forni Glacier, we observed a mean snow depth peak of 222 cm, ranging from 134 to 280 cm.

*- The English language of the paper is often odd and needs to be improved in revised version.*

Before uploading the reviewed manuscript, as suggested, the standard of English spelling and grammar will be improved by a professional, mother-tongue consultant

The following comments and corrections are the ones reported by the Referee in PDF of the paper.

*Title*: we will modify it accordingly.

*Line 14, Abstract:* We will deleted the word "snowfall" accordingly.

*Lines 15-18, Abstract:* We will delete the sentences accordingly.

*Line 19, Abstract: "fresh snow" Please use the terms defined in the international classification for seasonal snow on the ground.*

We used the terminology reported in the 2008 WMO Guide, where the term "fresh snow" is used instead of the term "new snow". In fact, the snowfall is defined as "the depth of freshly fallen snow deposited over a specified period (generally 24 h)". However, following the international classification for seasonal snow on the ground (Fierz et al., 2009), the height of new snow is "the depth in centimetres of freshly fallen snow that accumulated on a snow board during a standard observing period of 24 hours". Therefore, it seems that both "fresh" and "new" snow refer to the snow fallen in a period of 24 hours. We will change "fresh snow" with "new snow" in accordance with what the Referee asked for.

*Lines 21-22, Abstract: "The results indicate that the daily SR50 sonic ranger measures allow a rather good estimation of the SWE, and the provided snow pit data are available for defining the site mean value of fresh snow density." unclear.*

We will reword this sentence of the Abstract.

*Line 24, Abstract:* We will modify all the SWE values from "m" to "mm" accordingly.

*Lines 32-33: "The study of spatial and temporal variability of the water resource deriving from snow melt (i.e. Snow Water Equivalent, SWE) is very important for the estimation of the hydrological balance at catchment scale.." strange english.*

We will reword this sentence of the Abstract.

*Line 42:* We will add "(the largest valley glacier in Italy)" accordingly.

*Line 87:* The *impact of the fresh snow age is nowhere mentioned!*

As previously mentioned, we will add some discussion about the processes affecting snow during the time window of a day, corresponding to the period in which we quantify the daily differences in snow depth data used for deriving SWE values.

*Line 108: How do you measure only liquid? I guess you mean unheated precipitation gauge.*

We will add "(by means of an unheated precipitation gauge)" accordingly.

*Lines 111-112: Since there is no information on the web available about this pillow, please provide the information about the size.*

We will add some information regarding the snow pillow and the pressure sensor: "The AWS Forni SPICE is equipped with a snow pillow (Park Mechanical steel snow pillow, 150 x 120 x 1.5 cm) and a barometer (STS ATM.1ST) for measuring the snow water equivalent (Table 1, Beaumont, 1965)".

*Line 114:* We will add "the" and "to be able to determine the snow depth" accordingly.

*Line 121:* Following also the suggestions of the first Referee, we will modify with "provided".

*Lines 129-131: I guess you have to adjust it manually? How often was it moved upward again? How did the surface height (or ice thickness) change during this 12 years.*

We do not mean that we manually move the AWS. It leans on the surface of the glacier, it is not fixed into the ice. Therefore, during the ice ablation period, the AWS follows the lowering of the surface. Following also the suggestions of the first Referee, we will modify the sentence from "adjust" to "move together with". The mean annual ice thickness change is about 4 m per year.

*Lines 132-134: The radiation and wind conditions must have changed due to this dislocation, which may also impact the snow conditions?*

As the distance between the two sites is about 500 m, the difference in elevation is only 44 m and the aspect is very similar, we do not expect a noticeable impact of the site change on snow depth. However, we will add some discussions about the site move in the revised manuscript following also the suggestions of the first Referee.

*Line 146: Which time? You did not provide any information about the measurement frequency of the different sensors?*

We will add in the "Method" section: "Data points are sampled at 60-second intervals and averaged over a 60-minute time period for SR50 sonic ranger, wind sensor and barometer, over a 30-minute time period for the sensors recording air temperature, relative humidity, solar and infrared radiation, and liquid precipitation, and over a 10-minute time period for USH8 sonic ranger and snow pillow. All data are recorded in a flash memory card, including the basic distribution parameters (minimum, mean, maximum, and standard deviation values)."

*Line 146: What about the settling of the snow cover during this time?*

As far as settling is concerned, $\Delta h_{snow-pit-j}$ from eq. 2 (see answers to the main comments) would be higher if $\Delta h_{tj}$ were calculated considering a shorter interval than 24 hours. However, on one side this would not be possible because the sonic ranger data have too high error to consider hourly resolution, and on the other side, new snow is defined by WMO considering a 24-hour period. In our opinion, therefore, we do not need

considering settling as new snow defined by WMO does already include the settling that occurs in the period used to measure new snow.

*Line 149:* We will modify the sentence from "snow days" to "days with snowfalls".

*Lines 150-153: This procedure is not clear. Could you provide more details. It seems to be crucial for the paper!*

In the new version of the manuscript, we will modify this part of the "Method" section (following also the suggestions of the first Referee) in order to better clarify our procedure in estimating the density of the new snow ($\rho_{new\,snow}$) used for deriving SWE from snow depth data (see answers to the main comments).

*Line 149: Why is this value presented as a new result in the abstract, if it has already been found in previous studies. But none of these studies really demonstrated how exactly the value was determined.*

Following also the suggestions of the first Referee, we will modified this paragraph and the Abstract. In the revised manuscript, we will explain better how the new snow density is estimated (see answers to the main comments).

*Lines 162-163: How does this compare with neighboring stations?*

We will add a comparison with data from 10 stations spread over the Italian Alps: "These values are in agreement with findings over the Italian Alps in the period 1960–2009. In fact, Valt and Cianfarra (2010) reported a mean snow depth of 233 cm for the stations above 1500 m a.s.l."

*Line 162:* We will modify the sentence accordingly adding "second".

*Line 169: You don't know without independent validation! Small snowfalls are impossible to detect due to uncertainty of the snow depth sensors and since there is often concurrent settling even snowfalls larger the measurement uncertainty can not be detected. impossible to detect and*

Following also the suggestions of the first Referee, we will delete this paragraph regarding the comparison between Campbell and Sommer sensors.

*Lines 182-184:* We will delete this sentence accordingly.

*Lines 186-187: This is strange! Could you provide any explanation? Why should the measurement be correct afterwards?*

The results from the snow pillow are difficult to explain as this sensor has been working for only two winter seasons and then we are still in the process of testing. Analyzing data of the next years will allow a more robust

interpretation. However, we have searched for a possible explanation of this over-weighing accordingly and perhaps this error may be due to the configuration of the snow pillow.

*Lines 191-192: Why do you think the snow pillow working correctly before January 2015, but not before January 2016?*

We found that the valid readings from the snow pillow occur whenever there is a snow cover thicker than about 50 cm. Because of the observed inter-annual variability in snow depth values, the beginning of the valid readings from the snow pillow differs from one year to the next. In fact, in the accumulation period 2015/2016 this threshold is reached after January 2016.

*Line 199: I suggest to add these measurements in Fig. 6.*

We will add in the Figure the values measured by means of the snow weighing tube, accordingly. The new version of the figure is:

[Figure]

*Line 205-206: Improve english!*

Following also the suggestion of the first Refeee, we will modify this paragraph.

*Line 206:* We will add "of the density by" accordingly.

*Lines 207-208: Percentage difference instead of the absolute values would provide much more information*.

Following also the suggestion of the first Referee, we will add "corresponding to about 14% of the mean total cumulative SWE considering all hydrological years".

*Line 212: really tens and not ten?*

The ice glacier surface features a lot of different conditions: bare ice, ponds of different size and depth, presence of dust and fine or coarse debris that can be scattered over the surface or aggregated. This surface heterogeneity translates into a differential ablation. In fact, each material has a different value of albedo and heat transfer and then the below ice will receive a different amount of energy for melting. The result is a surface roughness of tens of centimeters.

*Lines 223-224: Not clear what you mean?*

Following also the suggestions of the first Referee, we will completely modify this paragraph.

*Lines 227-229: Which would not be true in your case since you measure over ice!*

Following also the suggestions of the first Referee, we will completely modify this paragraph.

*Line 232-246: What is this? Does not belong to Discussion and has not direct connection to your results.*

Following also the suggestions of the first Referee, we will move this part to the "Introduction" section.

*Lines 254-256: What is this temperature threshold for the AWS1?*

We will add accordingly "From the Forni Glacier, the application of the +0.5°C daily temperature threshold allows for a consistent quantification of snow ablation while, instead, for detecting the beginning of the snow melting processes a suitable threshold has proven to be at least −4.6°C."

*Line 260:* Following also the suggestions of the first Referee, we will modify this part from "The unique issue is represented by the definition of…" to "In spite of all these issues, the SR50 sonic ranger features the unique problem represented by the definition of…"

*Figures:* We will modify all the figures and captions accordingly.

We very much appreciate the time and effort you put into the comments.

Sincerely,

Antonella Senese and Co-author

---

## Author Response (AR1)

Dear Editor,

We revised the manuscript in accordance with the comments from the reviewers. We now feel that the paper is more clear and understandable. We wish to thank the reviewers for the helpful suggestions, which we believe strongly improved our manuscript.

Please find below the detailed comments and responses to the Reviewers' suggestions. In the revised manuscript, the changes are colored in yellow.

We hope that the revised manuscript can meet your and the Referees' expectations, and be accepted for publication on the journal; otherwise, we are open to new improvements.

Many thanks for your help.
Best regards,

Antonella Senese and co-authors

**Referee #1**

Dear Referee

Thanks for helping us in improving the manuscript.

We have considered all your comments in the revised version of the manuscript. The title, as you suggest and in agreement also with the second Referee, is: Estimating the snow water equivalent on remote and glacierized high elevation areas (Forni Glacier, Italy).

We have therefore modified the "Method" section in order to better explain the approach applied for estimating the site-average-new-snow density ($\rho_{new\ snow}$) and the sonic-ranger-depth-derived $SWE$. In the previous version of the manuscript, we performed the analyses using the mean new snow density (140 kg m$^{-3}$) that was obtained by Senese et al. (2012) considering the 2005-2009 dataset, and then we discussed to what degree this value is able to describe the data from the years to come. In the revised version of the manuscript, we have started with updating the site average new snow density estimation exploiting all the available datasets and we have performed all the subsequent analyses using this value.

The text concerning this issue in "Method" section is:

> In addition to the measurements recorded by the AWSs, since winter 2005-2006, personnel from the Centro Nivo-Meteorologico (namely CNM Bormio-ARPA Lombardia) of the Lombardy Regional Agency for the Environment have periodically used snow pits (performed according to the AINEVA protocol, see also Senese et al., 2014) in order to estimate snow depth and $SWE$. In particular, for each snow pit $j$, the thickness ($h_{ij}$) and the density ($\rho_{ij}$) of each snow layer ($i$) are measured for determining its snow water equivalent, and then the total $SWE_{snow-pit-j}$ of the whole snow cover ($n$ layers) is obtained:
>
> $$SWE_{snow-pit-j} = \sum_{i=1}^{n} h_{ij} \cdot \frac{\rho_{ij}}{\rho_{water}} \tag{1}$$
>
> where $\rho_{water}$ is water density. As noted in a previous study (Senese et al., 2014), the date when the snow pit is dug is very important for not underestimating the actual accumulation. For this reason, we considered only the snow pits excavated before the beginning of snow ablation. In fact, whenever ablation occurs, successive $SWE$ values derived from snow pits show a decreasing trend (i.e., they are affected by mass losses).
>
> The snow pit $SWE$ data were then used, together with the corresponding total new snow derived from sonic ranger readings, to estimate the site average $\rho_{new\ snow}$, in order to update the value of 140 kg m$^{-3}$ that was found in a previous study of data of the same site covering the 2005-2009 period (Senese et al., 2012a). We need to update our figures for $\rho_{new\ snow}$ as it is the key variable for estimating $SWE$ from the sonic ranger's new snow data. Specifically, for each snow pit $j$, the corresponding total new snow was first determined by:
>
> $$\Delta h_{snow-pit-j} = \sum_{t=1}^{m} (\Delta h_{tj}) \tag{2}$$
>
> where $m$ is the total number of days with snowfall in the period corresponding to snow pit $j$ and $\Delta h_{tj}$ corresponds to the depth of new snow on day $t$. In particular, we considered the hourly snow depth values recorded by the sonic ranger in a day and we calculated the difference between the last and the first reading. Whenever this difference is positive (at least 1 cm), it corresponds to a new snowfall. All data are subject to a strict control to avoid under- or over-measurements, to remove outliers and nonsense values, and to filter possible noises. $\sum_{t=1}^{m} (\Delta h_{tj})$ is therefore the total new snow measured by the Campbell SR50 from the beginning of the accumulation period to the date of the snow pit survey. The average site $\rho_{new\ snow}$ was then determined as:
>
> $$\rho_{new\ snow} = \frac{\sum_{j=1}^{k} SWE_{snow-pit-j}}{\sum_{j=1}^{k} (\Delta h_{snow-pit-j})} \tag{3}$$
>
> where $j$ identifies a given snow pit and the corresponding total new snow and the sum extends over all $k$ available snow pits. Instead of a mere average of $\rho_{new\ snow}$ values obtained from individual snow pit surveys, this relation gives more weight to snow pits with a higher $SWE_{snow-pit}$ amount.
>
> The $SWE$ of each day ($t$) was then estimated by:
>
> $$SWE_{SR-t} = \begin{cases} \Delta h_t \frac{\rho_{new\ snow}}{\rho_{water}} & if\ \Delta h_t \geq 1\ cm \\ 0 & if\ \Delta h_t < 1\ cm \end{cases} \tag{4}$$

We have also applied the leave-one-out cross-validation technique (LOOCV, a particular case of leave-p-out cross-validation with p = 1) to ensure independence between the data we use to estimate $\rho_{new\ snow}$ and the data we use to assess the corresponding estimation error. Specifically, we applied Eq. (3) (see answer above) once for

each snow pit ($j$), using all other snow pits in the calculation (*LOOCV* $\rho_{new\,snow}$) and using the selected snow pit as a single-item test ($\rho_{new\,snow}$ from snow pit $j$). In this way, we avoid dependence between the calibration and validation datasets in assessing the new snow density.

The results give evidence that the standard deviation of the differences between the *LOOCV* $\rho_{new\,snow}$ values and the corresponding single-item test values ($\rho_{new\,snow}$ from snow pit $j$) is 18 kg m$^{-3}$. The error of the average value of $\rho_{new\,snow}$ can therefore be estimated dividing this standard deviation for the square root of the number of the considered snow pits. It turns out to be 6 kg m$^{-3}$. The new and the old estimates of $\rho_{new\,snow}$ (149 and 140 kg m$^{-3}$, respectively) therefore do not have a statistically significant difference. The individual snow accumulation periods instead have a naturally higher error and the single snow pit estimates for $\rho_{new\,snow}$ range from 128 to 178 kg m$^{-3}$. In addition, we attempted to extend this analysis considering each single snow layer ($h_{ij}$) instead of each snow pit $j$. In particular, we tried to associate to each snow pit layer the corresponding new snow measured by the sonic ranger (Citterio et al., 2007). However, this approach turned out to be too subjective to contribute more quantitative information about the real representativeness of the ρnew snow value we found.

All these analyses are shown in the "Discussion" section of the new version of the manuscript.

Finally, before uploading the reviewed manuscript, as suggested by the second Referee, the standard of English spelling and grammar has been improved by a professional, mother-tongue consultant.

**Point-by-point answers to your comments follow:**

*Discuss the potential errors in individual events and how this impacts your peak estimation. A large event (big increase in snow depth) that has a large deviation from the mean density will result in larger errors (e.g. a heavy wet snowfall). What is the potential for this to occur at this site? Add a discussion about missing data as this is the greatest threat to failure of the technique. Can you do gap filling with photographed snow stakes? Would you recommend redundant sensors? More specific comments are listed below.*

We have addressed these issues in the "Discussion" section of the new version of the manuscript.
As far as large events are concerned, they are rather rare at the studied site: only 3 days in the 11-year period covered by the data recorded more than 40 cm of new snow (the number of days decreases to 1 if the threshold increases to 50 cm). More in detail, we have the following distribution of new snow: 382 days with values between 1 and 10 cm, 95 days with values between 10 and 20 cm, 33 days with values between 20 and 30 cm, 11 days with values between 30 and 40 cm. Beside investigating the distribution of new snow values, we have checked also if the days in the different new snow intervals have significantly different average temperatures. We have not found any signal.
We agree that missing data is a relevant issue. The introduction of the second sonic ranger (Sommer USH8) at the end of the snow season 2013-2014 was an attempt to reduce the impact of this problem. The second sonic ranger, however, was still in process of testing in the last years of the period investigated within this paper (we e.g. changed the sensor model). We are confident that in the years to come it can help reducing the problem of missing data. Multiple sensors for fail-safe data collection are indeed highly recommended.
The four stakes installed at the corners of the snow pillow at the beginning of the 2014-2015 snow season were another idea for collecting more data. Unfortunately, they were broken almost immediately after the beginning of the snow accumulation period. They can be another way to deal with the problem of missing data, provided we figure out how to avoid breakage during the winter season
Finally, we have attempted to extend the analysis considering each single snow layer ($h_{ij}$) instead of each snow pit $j$. In particular, we have tried to associate to each snow pit layer the corresponding new snow measured by

the sonic ranger (Citterio et al., 2007). However, this approach turned out to be too subjective to contribute more quantitative information about the real representativeness of the ρnew snow value we found.

*Note that the units for SWE should be reported in mm water equivalent (w.e.) or kg m-2 and not m w.e. . Snow depths should also be reported in cm and not m. It would also be useful if you used an abbreviation for "sonic ranger-derived SWE" such as SWE$_{SR}$ and use this throughout the paper.*

We have modified throughout the manuscript accordingly.

*Abstract*

*Page 1, Line 15: "…on the Forni Glacier in Italy."*

We have modified the sentence accordingly.

*Lines 19-20: This sentence misses the mark. From what I have read, you are not really assessing the mean value of new snow density…this was done elsewhere. I think you miss explicitly stating the aim of the analysis and the value of this paper. You should state here that you are using mean new snowfall density and automated depth measurements to estimate the SWE of new snowfall and accumulating to estimate peak SWE and evaluating against other methods.*

We have modified the sentence accordingly: "The aim of the analyses is to estimate the SWE of new snowfall and the annual peak of SWE based on the average density of the new snow at the site (corresponding to the snowfall during the standard observation period of 24 hours) and automated depth measurements, as well as to find the most appropriate method for evaluating SWE at this measuring site."

*Line 21: "rather good" is a vague and subjective description of the estimation. Avoid this and/or quantify the estimation.*

We have calculated the RMSE between SWE derived from SR50 sonic ranger and measured by means of snow pillow and we have found a RMSE of 45 mm w.e. We have added this information in the abstract accordingly.

*Introduction*

*Page 2, Line 37: "…often only snowfall measurements are available…"*

We have modified the sentence accordingly.

*Line 38: "assess" should be "calculate" and "depending" should be "depends". Fresh snow density also depends on surface conditions, correct?*

We have modified the sentences accordingly. We have added "surface conditions" in addition to "atmospheric conditions".

*Line 50: CryoNet is more of a network attached to the GCW initiative, rather than a "project"*

We have modified the sentence accordingly.

*Line 71: "detail" not "details"*

We have modified the sentence from "the accuracy should be verified in details for a large variety of events" to "the accuracy should be carefully verified for a large variety of events".

*Page 3, Lines 89-93: For item ii, I'm not sure that you are "defining the reliability of…" because you don't really have a solid reference (more on that later) to be able to do that. I would rather you said that you were "assessing the capability of…". You also use the term "obtained SWE" here…now I assume that you are referring to the derived SWE from the depth measurements so you need to be more specific here.*

We have modified the sentence accordingly: "ii) assess the capability to obtain SWE values from the depth measurements".

**Data and Methods**

*Page 3, Line 100: You refer to "These sensors" but you should rather say "These measurements were made at the two stations…" since not all of the measurements were made with sensors.*

We have modified the sentence accordingly.

*Line 106: Was the T/RH sensor shielded? I assume yes.*

The thermo-hygrometer is with shield. We have modified the sentence accordingly adding "shielded".

*Page 4, Line 110: You should cite Beaumont (1965) here when introducing the snow pillow methods. Also some grammar issues with the first sentence on this page.*

We have modified the sentence accordingly citing Beaumont (1965) and checking the grammar issues. The new version is: "AWS Forni SPICE is equipped with a snow pillow (Park Mechanical steel snow pillow, 150 x 120 x 1.5 cm) and a barometer (STS ATM.1ST) for measuring the snow water equivalent (Table 1, Beaumont, 1965). "

*Line 116: I think "constrictions" should be "challenges".*

We have modified the sentence accordingly.

*Line 121: "represented" should be "provided" or "supplied". "deep" should be "thorough".*

We have modified the sentences accordingly.

*Line 122: "working" should perhaps be "operation".*

We have modified the sentence accordingly.

*Line 123: "…due to ice flow, etc.) is required before installation…"*

We have modified the sentence accordingly.

*Lines 125-128: Not sure if this paragraph adds anything to the paper*

We have deleted this paragraph accordingly.

*Line 130: I think that I know what you mean by "adjust" but you should clarify this.*

We have modified the sentence accordingly in order to better explain what we mean. The new version is : "In this way, the AWSs stand freely on the ice, and move together with the melting surface during summer (with a mean ice thickness variation of about 4 m per year)."

*Line 132: What are ring faults?*

The ring faults are a series of circular or semicircular fractures with stepwise subsidence, caused by englacial or subglacial meltwater creating voids at the ice-bedrock interface and eventually the collapse of cavity roofs. We have modified the paragraph accordingly. The new version is: "Due to the formation of ring faults, in November 2015 both AWSs were moved to the Forni glacier central tongue (46°23'42.40"N and 10°35'24.20"E at an elevation of 2675 m a.s.l., the red star in Fig. 1a). Ring faults are a series of circular or semicircular fractures with stepwise subsidence (caused by englacial or subglacial meltwater) that could compromise the stability of the stations because they could create voids at the ice-bedrock interface and eventually the collapse of cavity roofs (Azzoni et al., 2017; Fugazza et al., submitted)."

*Line 146-147: "…from snow depth acquired by sonic rangers and estimated new snowfall density." Clarify what you mean by the last sentence "In particular…". This is a crucial piece of the methodology and you should describe this better. What time did you use for start/end? Did you do any noise filtering? How do you account for redistribution, settling, and sublimation during the day? (should also be discussed in the Discussion section).*

In order to estimate the "daily positive differences in depth ($\Delta h$)", we considered the hourly snow depth values recorded by the sonic ranger in a day and we calculated the difference between the last and the first reading. Whenever this difference is positive (at least 1 cm), it corresponds to a new snowfall:

$$\Delta h = \begin{cases} reading\ at\ h23 - reading\ at\ h00\ (if\ positive\ and\ > 1\ cm) \\ 0\ (if\ negative\ or\ \leq 1\ cm) \end{cases}$$

This issue is better explained in the revised version of the manuscript (see answers to the main comments).

As far as settling is concerned, $\Delta h_{snow-pit-j}$ from Eq. 2 (see answers to the main comments) would indeed be higher if $\Delta h_{tj}$ values were calculated considering an interval shorter than 24 hours. However, this would not be possible because on the one hand, the sonic ranger data's margin of error is too high to consider hourly resolution, and on the other hand, new snow is defined by the WMO within the context of a 24-hour period. Therefore, settling could not be considered in our analyses since new snow as defined by the WMO already includes the settling that occurs in the 24-hour period.

Obviously, our sonic ranger data may be affected by snow transported by wind. The effect that is potentially more relevant is new snow that is recorded by the sonic ranger but then blown away in the following days. It is therefore considered in $\Delta h_{snow-pit-j}$ but not in $SWE_{snow-pit-j}$, thus causing an underestimation of $\rho_{new\ snow}$ (see Eq. 3 – answers to the main comments). The snow transported to the measuring site can also influence $\rho_{new\ snow}$ even if in this case the effect is less important as it measured both by the sonic ranger and by the snow pit. Here, the problem may be an overestimation of $\rho_{new\ snow}$ as snow transported by wind usually has a higher density than new snow. We considered the problem of the effect of wind on snow cover when we selected the

station site on the glacier. Even though sites not affected by wind transport simply do not exist, we are confident that the site we selected has a position that can reasonably minimize this issue.

Moreover, sublimation processes would have an effect that is similar to those produced by new snow that is recorded by the sonic ranger but then blown away in the following days. In any case, the value we found for the site average new snow density (i.e. 149 kg m$^{-3}$) does not seem to suggest an underestimated value.

All these eventual effects are described in the "Discussion" section.

As regards the quality-check activities, we submitted the dataset to a strict control to avoid under- or over-measurements, to remove outliers and nonsense values, and to filter possible noises. We will modify the relative paragraph in the "Method" section accordingly in order to better explain our approach (see answer to the main comments).

*Page 5, Lines 152-153: What do you mean by "unique date"?*

We have modified the sentence accordingly in order to better clarify the approach (see answer to the main comments).

*Lines 150-153: This is the methodology that isn't clearly described (see note above).*

In the new version of the manuscript, we have modified this part of the "Method" section (following also the suggestions of the second Referee) in order to better clarify our procedure in estimating the density of the new snow ($\rho_{new\ snow}$) used for deriving SWE from snow depth data (see answers to the main comments).

*Results*

*Beside expressing the errors or biases in absolute units, it would be helpful if the relative bias was stated. Same can be said for subsequent sections.*

We have modified all the showed values adding the error accordingly.

*Lines 159-161: Can you comment here or in the discussion on any potential impacts on the analysis due to the site move.*

The relocation of the AWSs could influence snow conditions, as there could be a different snow accumulation due to a different radiation input and diverse wind regimes. However, as the distance between the two sites is about 500 m, the difference in elevation is only 44 m and the aspect is very similar, we do not expect a noticeable impact of the site change on snow depth.

*Lines 166-169: The intercomparison between the SR50 and the USH8 is interesting but largely irrelevant for this paper. I would omit this. What you could mention is that the two instruments had a correlation of ?? and that from 2015 onwards, redundancy in the snow depth measurement could mean better data for the SWE estimate.*

We have omitted the intercomparison and the relative figure accordingly. We have moved this part into the "Method" section, specifying the correlation between the datasets from Campbell and Sommer sensors.

*Line 174: What is "a very good agreement"?*

We have added the root mean square error, that is equal to 58 mm w.e.

*Line 180: "elaborate" should be "accumulate"*

We have modified the sentence accordingly.

*Figure 4: it would be very useful if the missing SR50 data could be indicated (e.g. different colour line, etc). This would certainly help with the interpretation of the graph. One might argue that all of the estimated SWE should be set to missing after large gaps in the snow depth data since it is an accumulated result.*

We have modified the figure showing the period without snow depth data. The new version of the figure is:

[Figure]

*Page 6, Line 188: "…thus suggesting a correct working of the sensor." This is more than a sensor, but rather a technique or process. This sentence is awkward and should be reworded.*

We have modified the sentence accordingly. The new version is: "thus suggesting that in spite of the problems at the beginning of the snow season, the snow pillow seems to give reasonable results"

*Line 192: typo "derided".*

We have modified in "$SWE_{SR}$" according to your previous comment.

*Line 194: Change "raises" to "increases".*

We have modified the sentence. The new version is: "There is a general underestimation of $SWE_{SR}$ compared to the snow pillow values, considering the 2014-2015 data, though the agreement strengthens in the 2015-2016 dataset (Fig. 5): 54 mm w.e. and 29 mm w.e. of RMSE regarding 2014-2015 and 2015-2016, respectively."

*Lines 196-197: You mention the snow course data using the snow tube and suggest a "large spatial variability". Perhaps you could report the average and standard deviation of this data and in the discussion, relate this variability to the differences you see between the SR50 SWE estimation and the snow pillow and snow pit.*

We have modified accordingly adding the mean value of 165 cm and the standard deviation of 29 cm. In addition, as reported in the following comment, we have discussed this variability in the "Discussion" section.

*Lines 200-201: Oversampling of the tube is a potential error and should be included in the discussion and not in the results. You need to estimate this error and put it into context with the differences between the methods.*

As stated in Johnson et al. (2015), numerous studies have been conducted to verify snow tube accuracy in determining SWE (Freeman, 1965; Work et al., 1965; Beaumont, 1967; Peterson and Brown, 1975; Goodison, 1978; Farnes et al., 1982), the most recent being by Sturm et al. (2010) and Dixon and Boon (2012). The most recent comparison of snow tubes by Dixon and Boon (2012) examined the Standard Federal, Meteorological Service of Canada, and SnowHydro snow tubes. The earlier studies had found that snow tubes tended to over-measure SWE from about 4–11%, whereas the recent studies by Sturm et al. (2010) and Dixon and Boon (2012) found that snow tubes could under-measure or over-measure SWE from -9% to 11%. Even if we allow for ±10% margin of error in our snow tube measurements, the high SWE variability is confirmed.

This part is discussed in the "Discussion" section of the revised manuscript.

**Discussion**

*The start of the Discussion paragraph needs a couple of sentences of introduction to "frame" the problem and set up the following paragraphs.*

We have added a paragraph accordingly in order to better introduce the "Discussion" section: "Defining a correct algorithm for modeling SWE data is very important for evaluating the water resources deriving from snow melt. The approach applied to derive $SWE_{SR}$ is highly sensitive to the value used for the new snow density, which can vary substantially depending on both atmospheric and surface conditions. In this way, the error in individual snowfall events could be quite large. Moreover, the technique depends on determining snowfall events, which are estimated from changes in snow depth, and the subsequent calculation and accumulation of $SWE_{SR}$ from those events. Therefore, missed events due to gaps in snow depth data could invalidate the calculation of peak $SWE_{SR}$. For these reasons, we focused our analyses on understanding how an incorrect assessment of ρnew snow or a gap in snow depth data may affect the estimation of the SWE."

*Lines 205-208: The whole success of technique hinges on the approximation of the average new snowfall density but yet this only gets a few sentences in the discussion. This needs to be expanded. When would you expect errors to be greatest? At this site, can you potentially get large snowfalls that have much greater (or lower) densities*

*than the average, which could then bias your accumulated SWE? Any suggestions on how to better estimate new snowfall SWE, perhaps at sites where you don't have snow pits to back-calculate the density.*

We have modified this paragraph accordingly. The new version is:

[revised manuscript text omitted]

*Line 205: "Once our procedure was verified, we performed…"*

We have modified this paragraph accordingly to the previous comment.

*Line 209: What is a "general good agreement". Quantify this.*

We have modified the sentence accordingly from "Beside to a general good agreement between the measures performed with the different sensors, there are also some problems." to "As regards the instrumentation, we found some issues related to the derived snow information."

*Line 214: "not constant" should be "inconsistent"*

We have modified the sentence accordingly.

*Page 7, Lines 223-231: You reference some issues with the snow pillow and then state that your snow pillow only seems to be working with depths greater that 50cm. Can you relate any of the referenced potential issues to the errors that you are seeing at the site? I'm not convinced that your issues are related to those referenced so this discussion needs to be stronger.*

We have modified this part accordingly. The new version is: "The results from the snow pillow are difficult to explain as this sensor has been working for only two winter seasons and we are still in the process of testing. Analyzing data from the years to come will allow a more robust interpretation. However, we have searched for a possible explanation of this problem and this error could be due to the configuration of the snow pillow."

*Lines 232-246: This paragraph is more of a justification for using this technique and is more appropriate for the introduction than the discussion. The sentences (Lines 246-250) about the snow pillow is relevant, however. The subsequent discussion about the snow pit should have its own paragraph.*

We have modified this paragraph accordingly, moving part of it into the "Introduction" section. In addition, we have discussed about the snow pit measurements in a separate paragraph.

*Line 248-249: You should refer to Fig 5 and discuss this error in context with what you are seeing at the site. This should be combined with the snow pillow discussion. Also, is there a reference for this statement about minimum snow pillow measurements?*

We have modified this section as described in the previous comments.

*Line 252: What do you mean by "whole glacier accumulation amount"?*

We have modified the sentence accordingly from "whole glacier accumulation amount" to "total snow accumulation amount".

*Line 253: "…the peak cumulative SWE…" Same applies for the next sentence.*

We have modified the sentences accordingly.

*Line 257: Typo "closed" should be "close"*

We have modified the sentence accordingly.

*Lines 259-261: The two sentences beginning with "Finally…" seem out of place or awkward. Please re-organize these.*

We have modified the sentences accordingly from "Finally, with data acquired by the SR50 sonic ranger a correct curve of SWE was derived. The unique issue is represented by the definition of the beginning of the accumulation period, but this can be overcome using albedo data." to "Finally, the SR50 sonic ranger features the unique problem of the definition of the start of the accumulation period, but this can be overcome using albedo data."

*Page 8, Lines 261-263: The discussion about the SR50/USH8 is largely unsubstantiated and should be omitted.*

We have deleted the discussion regarding the USH8 sonic ranger accordingly.

*I would like to see me recommendations in the discussion about how to make this technique better and how to adapt it for other remote sites. Perhaps make some recommendations on further testing, possibly to take advantage of other data collected during SPICE.*

We have modified the "Discussion" section accordingly, following all your previous suggestions and comments.

**Conclusions**

*Line 267: Was the acronym SPICE not defined earlier in the paper? If it was, it doesn't need to be defined here.*

We have deleted the definition of the acronym accordingly.

*Line 269: The sentence beginning with "This has allowed…" needs some commas to read correctly.*

We have modified the sentence accordingly. The new version is: "This has allowed an accurate comparison and evaluation of the pros and cons of using the snow pillow, sonic ranger, snow pit, or snow weighing tube, and of estimating SWE from snow depth data."

*Line 270: "…SWE values using a fresh snow density…"*

We have modified the sentence accordingly.

*Lines 270-273: I have some problems with the statement "The results achieved…". You only tested two sensors for measuring snow depth so I wouldn't go as far as saying that the SR50 is the "most suitable device"…perhaps it is but you don't have enough information to substantiate this. You present a technique for estimating peak SWE on a glacier and you compare this technique against another technique (e.g. the snow pillow) but you don't really have a true reference so I'm not sure you can say that it is "most suitable". Rather, I would reference your relative error and despite the issues, suggest that the technique is "suitable" for estimating SWE.*

We have modified the sentences accordingly from "The results achieved during the SPICE experiment support our procedure for deriving SWE values and the applied fresh snow density of 140 kg m$^{-3}$ (Senese et al., 2014),

and suggest that, once $\rho_{fresh}$ snow is known, the SR50 sonic ranger can be considered the most suitable device on a glacier to record snowfall events and to measure snow depth values in order to derive the point SWE."

To "We found that the mean new snow density changes based on the considered period was: 140 kg m$^{-3}$ in 2005-2009 (Senese et al., 2014) and 149 kg m$^{-3}$ in 2005-2015. The difference is however not statistically significant. We first evaluated the new snow density estimation by means of LOOCV and we found an error of 6 kg m$^{-3}$. Then, we benchmarked the derived $SWE_{SR}$ data against the information from the snow pillow (data which was not used as input in our density estimation), finding a RMSE of 45 mm w.e. These analyses permitted a correct definition of the reliability of our method in deriving $SWE$ from snow depth data. Moreover, in order to define the effects and impacts of an incorrect $\rho_{new\,snow}$ value in the derived $SWE$ amount, we found that a change in density of ±25 kg m$^{-3}$ causes a mean variation of 17% of the mean total cumulative $SWE$ considering all hydrological years. Finally, once $\rho_{new\,snow}$ is known, the sonic ranger can be considered a suitable device on a glacier, or in a remote area in general, for recording snowfall events and for measuring snow depth values in order to derive $SWE$ values. In fact, the methodology we presented can be interesting for other sites as it allows estimating total $SWE$ using a relatively inexpensive, low power, low maintenance, and reliable instrument such as the sonic ranger, and it is a good solution for estimating $SWE$ at remote locations such as glacier or high alpine regions."

*Line 274: What are the relative variations as a result of the density change?*

We have modified the sentence accordingly adding the relative variation: "Moreover, in order to define the effects and impacts of an incorrect $\rho_{new\,snow}$ value in the derived $SWE$ amount, we found that a change in density of ±25 kg m$^{-3}$ causes a mean variation of 17% of the mean total cumulative $SWE$ considering all hydrological years."

*Instead of ending the conclusions with a comment about the error, perhaps end with a comment on the applicability to other remote sites.*

We have added accordingly: "The sensors generally used (e.g. heated tipping bucket rain gauges, heated weighing gauges, or disdrometers) can provide more accurate measurements compared to the ones installed at the Forni Glacier. The problem is that in remote areas like a glacier at a high alpine site, it is very difficult to install and maintain them. The main constrictions concern i) the power supply to the instruments, which consists in solar panels and lead-gel batteries, and ii) the glacier dynamics, snow flux and differential snow/ice ablation that can compromise the stability of the instrument structure. Therefore, for our limited experience in such remote areas, a sonic ranger could represent a useful approach for estimating SWE, since it does not require expert personnel, nor does it depend on the date of the survey (as do such manual techniques as snow pits and snow weighing tubes); it is not subject to glacier dynamics, snow flux or differential ablation (as do graduated rods installed close to an automated camera and snow pillows), and the required power is not so high (unlike heated tipping bucket rain gauges). The average new snow density must, however, be known either by means of snow pit measurements or by the availability of information from similar sites in the same geographic area."

We very much appreciate the time and effort you put into the comments.

Sincerely,

Antonella Senese and Co-author

**Referee #2**

Dear Referee

Thanks for helping us in improving the manuscript.

We have considered all your comments in the revised version of the manuscript. The title, as you suggest and in agreement also with the second Referee, is: Estimating the snow water equivalent on remote and glacierized high elevation areas (Forni Glacier, Italy).

We have therefore modified the "Method" section in order to better explain the approach applied for estimating the site-average-new-snow density ($\rho_{new\ snow}$) and the sonic-ranger-depth-derived *SWE*. In the previous version of the manuscript, we performed the analyses using the mean new snow density (140 kg m$^{-3}$) that was obtained by Senese et al. (2012) considering the 2005-2009 dataset, and then we discussed to what degree this value is able to describe the data from the years to come. In the revised version of the manuscript, we have started with updating the site average new snow density estimation exploiting all the available datasets and we have performed all the subsequent analyses using this value.

The text concerning this issue in "Method" section is:

[revised manuscript text omitted]

All these analyses are shown in the "Discussion" section of the new version of the manuscript.

Finally, before uploading the reviewed manuscript, as suggested by the second Referee, the standard of English spelling and grammar has been improved by a professional, mother-tongue consultant.

**Point-by-point answers to your comments follow.**

*- the exact determination of the fixed density*

In the new version of the manuscript, we have modified this part of the "Method" section (following also the suggestions of the first Referee) in order to better clarify our procedure in estimating the density of the average new snow ($\rho_{new\,snow}$) used for deriving SWE from snow depth data (see answers to the main comments).

*- the calculation of the daily positive snow depth differences*

In order to estimate the "daily positive differences in depth ($\Delta h$)", we considered the hourly snow depth values recorded by the sonic ranger in a day and we calculated the difference between the last and the first reading. Whenever this difference is positive (at least 1 cm), it corresponds to a new snowfall:

$$\Delta h = \begin{cases} reading\ at\ h23 - reading\ at\ h00\ (if\ positive\ and\ > 1\ cm) \\ 0\ (if\ negative\ or\ \leq 1\ cm) \end{cases}$$

This issue is better explained in the revised version of the manuscript (see answers to the main comments).

*- the limitations and uncertainty of the method*

For validating our procedure in estimating the new snow density, we have applied the leave-one-out cross-validation technique (LOOCV, a particular case of leave-p-out cross-validation with p = 1) in order to assess both the error of the estimation of the average site $\rho_{new\,snow}$ and the error we perform if we estimate $\rho_{new\,snow}$ of each single snow pit by means of our approach (see answers to the main comments).

In addition, we have investigated the *SWE* sensitivity to changes in $\rho_{new\,snow}$: we have calculated *SWE$_{SR}$* using different values of new snow density ranging from 100 to 200 kg m$^{-3}$ at 25 kg m$^{-3}$ intervals.

We have also added to the "Discussion" section some new information on the occurrence of outliers in our data set (the highest new snow value is slightly higher than 40 cm) and on the distribution of new snow. We have then underlined that we do not have a significant link between new snow and corresponding daily average temperature that could e.g. indicate that higher new snow values are more frequent at the beginning and at the end of the snow season.

Moreover, we have investigated if potential errors in individual snowfall events could affect peak $SWE_{SR}$ estimation. In fact, a large snowfall event with a considerable deviation from the mean new snow density will result in large errors (e.g. a heavy wet snowfall). These events are rather rare at the Forni site: only 3 days in the 11-year period covered by the data recorded more than 40 cm of new snow (the number of days decreases to 1 if the threshold increases to 50 cm). More in detail, we found the following distribution of new snow: 382 days with values between 1 and 10 cm, 95 days with values between 10 and 20 cm, 33 days with values between 20 and 30 cm, 11 days with values between 30 and 40 cm. Beside investigating the distribution of new snow values, we also checked if the days in the different new snow intervals have significantly different average temperatures. The calculated average temperature values are -5.7 ± 4.5°C, -5.2 ± 4.2°C, and -4.8 ± 3.2°C (for days with 1-10 cm, 10-20 cm, and >20 cm of new snow depth, respectively), suggesting that there is no significant change of air temperature in these three classes.

Another possible source of error in estimating new snow density and in deriving daily SWE is represented by rainfall events. In fact, one of the effects is an enhanced snow melt and then a decrease in snow depth, as rain water has a higher temperature than the snow. Therefore, especially at the beginning of the snow accumulation season, we could detect a snowfall (analyzing snow depth data) but, whenever it was followed by a rainfall, the new fallen snow could partially or completely melt, thus remaining undetected when measured at the end of the accumulation season using snow pit techniques. This is therefore another potential error that, besides the ones previously considered, could lead to underestimation of the $\rho_{new\ snow}$ value, even if, as already mentioned, the found value of 149 kg m$^{-3}$ does not seem to suggest this. On the other hand, rain can also increase the SWE measured by the snow pit without giving a corresponding sign in the snow depth measured by the sonic ranger whenever limited amounts of rain fall over cold snow. Anyway, rain events are extremely rare during the snow accumulation period.

In addition to the validation of new snow density estimation, we have benchmarked the derived SWE data against the ones measured by the snow pillow (data not used as input in our density estimation). This validation permitted to correctly define the reliability of our method in deriving SWE from snow depth data.

All these data elaborations and analyses are included in the "Discussion" section.

*- transferability to other sites and climates*

In our opinion, the methodology we presented can be interesting for other sites as it allows estimating total SWE using a relatively inexpensive, low power, low maintenance, and reliable instrument such as the sonic ranger, and it is a good solution for estimating SWE at remote locations such as glacier or high alpine regions. We have highlighted this point in the "Conclusions" of the revised manuscript.

*The fixed density of 140 kg/m3 has already been used in earlier papers (2012, 2104) from you. Indeed the comparison between the estimated SWE from snow depth measurements and SWE from snow pits (the current Fig. 4) has already been published in The Cryosphere in 2014 (Fig. 2 in https://www.the-cryosphere.net/8/1921/2014/tc-8-1921-2014.pdf). However, there the estimated SWE data seem to be different from the one shown in the current paper? Can you explain?*

The values shown in the previous paper are the same but cumulating daily SWE data the yearly end date is different. In this study, we showed the SWE values only during the snow accumulation period, neglecting the ice ablation period. In fact, in this period the fallen snow is completely melted within a few days and always before the beginning of the accumulation period. Instead, in Senese et al. (2014) we cumulated daily SWE values until 30[th] September of each year. Only for this reason, the two figures seem different, but they show the same datasets.

However, in the revised manuscript, we have shown SWE data derived using the update new snow density (149 kg m[-3], instead of 140 kg m[-3])

*- Your fixed density is not a fresh snow density, since you totally neglect settling and are not able to determine small snow falls due to the measurement uncertainty of the snow depth sensor. The found density of 140 kg/m3 can therefore not be compared with published fresh snow densities found in literature and is relatively large since it has to compensate the missing snowfall amounts mentioned above.*

We have modified our terminology accordingly from "fresh snow density" to "new snow density". In addition, we agree that missing data is a relevant issue.
The introduction of the second sonic ranger (Sommer USH8) at the end of the 2013-2014 snow season was an attempt to reduce the impact of this problem. The second sonic ranger, however, was still in process of testing in the last years of the period investigated within this paper. We are confident that in the years to come it can help reduce the problem of missing data. Multiple sensors for fail-safe data collection are indeed highly recommended. In addition, the four stakes installed at the corners of the snow pillow at the beginning of the 2014-2015 snow season were another idea for collecting more data. Unfortunately, they were broken almost immediately after the beginning of the snow accumulation period. They can be another way to deal with the problem of missing data, provided we figure out how to avoid breakage during the winter season.

*- The impact of rain events at the beginning and end of the snow season has not been discussed so far.*

We have added some discussion about the impact of rain events: another possible source of error in estimating new snow density and in deriving the daily *SWE* is represented by rainfall events. In fact, one of the effects is an enhanced snow melt and then a decrease in snow depth, as rain water has a higher temperature than the snow. Therefore, especially at the beginning of the snow accumulation season, we could detect a snowfall (analyzing snow depth data) but, whenever it was followed by a rainfall, the new fallen snow could partially or completely melt, thus remaining undetected when measured at the end of the accumulation season using snow pit techniques. This is therefore another potential error that, besides the ones previously considered, could lead to underestimation of the $\rho_{new\ snow}$ value, even if, as already mentioned, the found value of 149 kg m[-3] does not seem to suggest this. On the other hand, rain can also increase the *SWE* measured by the snow pit without giving a corresponding sign in the snow depth measured by the sonic ranger whenever limited amounts of rain fall over cold snow. Anyway, rain events are extremely rare during the snow accumulation period."

*- Since your focus is the determination SWE from snow depth your title needs to be changed to something like: Estimation of SWE from automatic snow depth measurements during accumulation on a high alpine glacier.*

Following also the suggestions of the first Referee, we have modified the title accordingly from "Snow data intercomparison on remote and glacierized high elevation areas (Forni Glacier, Italy)" to "Estimating the snow water equivalent on remote and glacierized high elevation areas (Forni Glacier, Italy)".

*- The content of the abstract is odd. The abstract needs to be rewritten.*

Following also the suggestions of the first Referee, we have modified the abstract accordingly. The new version is:

"We present and compare 11 years of snow data (snow depth and snow water equivalent, *SWE*) measured by an Automatic Weather Station corroborated by data resulting from field campaigns on the Forni Glacier in Italy. The aim of the analyses is to estimate the *SWE* of new snowfall and the annual peak of *SWE* based on the average density of the new snow at the site (corresponding to the snowfall during the standard observation period of 24 hours) and automated depth measurements, as well as to find the most appropriate method for evaluating *SWE* at this measuring site. The results indicate that the daily SR50 sonic ranger measures allow a rather good estimation of the *SWE* (RMSE of 45 mm w.e. if compared with snow pillow measurements), and the available snow pit data can be used to define the mean new snow density value at the site. For the Forni Glacier measuring site, this value was found to be $149 \pm 6$ kg m$^{-3}$. The *SWE* derived from sonic ranger data is quite sensitive to this value: a change in new snow density of $\pm 25$ kg m$^{-3}$ causes a mean variation in *SWE* of $\pm 106$ mm w.e. for each hydrological year (corresponding to about 17% of the mean total cumulative *SWE* considering all hydrological years), ranging from $\pm 43$ mm w.e. to $\pm 144$ mm w.e.."

*- The possible impact of the dislocation of the station needs to be discussed.*

The relocation of the AWSs could influence snow conditions, as there could be a different snow accumulation due to a different radiation input and diverse wind regimes. However, as the distance between the two sites is about 500 m, the difference in elevation is only 44 m and the aspect is very similar, we do not expect a noticeable impact of the site change on snow depth.

*- You need to present some numbers about the uncertainty of the involved sensors, manual measurements and the uncertainty of the presented results. That means the papers definitely needs more quantitative information about the performance of your method.*

Following also the suggestions of the first Referee, we have added and discussed the uncertainty of the method for deriving the new snow density and of all the measurements and techniques applied in this study for quantifying *SWE*:
- The average site new snow density is found to be affected by an error of $\pm 6$ kg m$^{-3}$ (by means of the leave-one-out cross-validation).
- The comparison between SWE derived by snow depth data and SWE measured by means of the snow pillow showed a RMSE of 45 mm w.e.
- Snow tubes could under-measure or over-measure SWE of about $\pm 10$% (as found by Sturm et al., 2010, and Dixon and Boon, 2012).

*- Information about the measurement frequency and aggregation of the data shown in the figures need to included.*

Data points are sampled at 60-second intervals and averaged over a 60-minute time period for the SR50 sonic ranger, wind sensor and barometer, over a 30-minute time period for the sensors recording air temperature, relative humidity, solar and infrared radiation, and liquid precipitation, and over a 10-minute time period for the USH8 sonic ranger and snow pillow. All data are recorded in a flash memory card, including the basic distribution parameters (minimum, mean, maximum, and standard deviation values). We have added all these information in the revised manuscript accordingly.

In addition, we have added in all the figure captions the aggregation of data.

*- A comparison with data from other sites is needed in order to be able to judge the usefulness of the method.*

Comparing our results with snow depth data from 10 stations spread over the Italian Alps, the snow depth peaks are in agreement with findings over the Italian Alps. In fact, analyzing data in the period 1960–2009 from 10 stations above 1500 m a.s.l. Valt and Cianfarra (2010) reported a mean snow depth of 233 cm (from 199 to 280 cm). At the Forni Glacier, we observed a mean snow depth peak of 222 cm, ranging from 134 to 280 cm. We have added this comparison in the "Results" section.

*- The English language of the paper is often odd and needs to be improved in revised version.*

Before uploading the reviewed manuscript, as suggested, the standard of English spelling and grammar has been improved by a professional, mother-tongue consultant.

**The following comments and corrections are the ones reported by the Referee in PDF of the paper.**

*Title*: We have modified it accordingly.

*Line 14, Abstract:* We have deleted the word "snowfall" accordingly.

*Lines 15-18, Abstract:* We have deleted the sentences accordingly.

*Line 19, Abstract: "fresh snow" Please use the terms defined in the international classification for seasonal snow on the ground.*

We used the terminology reported in the 2008 WMO Guide, where the term "fresh snow" is used instead of the term "new snow". In fact, the snowfall is defined as "the depth of freshly fallen snow deposited over a specified period (generally 24 h)". However, following the international classification for seasonal snow on the ground (Fierz et al., 2009), the height of new snow is "the depth in centimetres of freshly fallen snow that accumulated on a snow board during a standard observing period of 24 hours". Therefore, it seems that both "fresh" and "new" snow refer to the snow fallen in a period of 24 hours. We have changed "fresh snow" with "new snow" in accordance with what the Referee asked for.

*Lines 21-22, Abstract: "The results indicate that the daily SR50 sonic ranger measures allow a rather good estimation of the SWE, and the provided snow pit data are available for defining the site mean value of fresh snow density." unclear.*

We have reworded this sentence of the Abstract.

*Line 24, Abstract:* We have modified all the SWE values from "m" to "mm" accordingly.

*Lines 32-33: "The study of spatial and temporal variability of the water resource deriving from snow melt (i.e. Snow Water Equivalent, SWE) is very important for the estimation of the hydrological balance at catchment scale.." strange english.*

We have reworded this sentence of the Abstract.

*Line 42:* We have added "(the largest valley glacier in Italy)" accordingly.

*Line 87:* The *impact of the fresh snow age is nowhere mentioned!*

As previously mentioned, we have added some discussion about the processes affecting snow during the time window of a day, corresponding to the period in which we quantify the daily differences in snow depth data used for deriving SWE values.

*Line 108: How do you measure only liquid? I guess you mean unheated precipitation gauge.*

We have added "(by means of an unheated precipitation gauge)" accordingly.

*Lines 111-112: Since there is no information on the web available about this pillow, please provide the information about the size.*

We have added some information regarding the snow pillow and the pressure sensor: "AWS Forni SPICE is equipped with a snow pillow (Park Mechanical steel snow pillow, 150 x 120 x 1.5 cm) and a barometer (STS ATM.1ST) for measuring the snow water equivalent (Table 1, Beaumont, 1965)."

*Line 114:* We have added "the" and "in order to observe the snow depth" accordingly.

*Line 121:* Following also the suggestions of the first Referee, we have modified with "provided".

*Lines 129-131: I guess you have to adjust it manually? How often was it moved upward again? How did the surface height (or ice thickness) change during this 12 years.*

We do not mean that we manually move the AWS. It leans on the surface of the glacier, it is not fixed into the ice. Therefore, during the ice ablation period, the AWS follows the lowering of the surface. Following also the suggestions of the first Referee, we have modified the sentence from "adjust" to "move together with". The mean annual ice thickness change is about 4 m per year.

*Lines 132-134: The radiation and wind conditions must have changed due to this dislocation, which may also impact the snow conditions?*

As the distance between the two sites is about 500 m, the difference in elevation is only 44 m and the aspect is very similar, so we do not expect a noticeable impact of the site change on snow depth.

*Line 146: Which time? You did not provide any information about the measurement frequency of the different sensors?*

We have added in the "Method" section: "Data points are sampled at 60-second intervals and averaged over a 60-minute time period for the SR50 sonic ranger, wind sensor and barometer, over a 30-minute time period for the sensors recording air temperature, relative humidity, solar and infrared radiation, and liquid precipitation, and over a 10-minute time period for the USH8 sonic ranger and snow pillow. All data are recorded in a flash memory card, including the basic distribution parameters (minimum, mean, maximum, and standard deviation values)."

*Line 146: What about the settling of the snow cover during this time?*

As far as settling is concerned, $\Delta h_{snow-pit-j}$ from eq. 2 (see answers to the main comments) would indeed be higher if $\Delta h_{tj}$ values were calculated considering an interval shorter than 24 hours. However, this would not be possible because on the one hand, the sonic ranger data's margin of error is too high to consider hourly resolution, and on the other hand, new snow is defined by the WMO within the context of a 24-hour period. Therefore, settling could not be considered in our analyses since new snow as defined by the WMO already includes the settling that occurs in the 24-hour period.

*Line 149:* We have modified the sentence from "snow days" to "days with snowfalls".

*Lines 150-153: This procedure is not clear. Could you provide more details. It seems to be crucial for the paper!*

In the new version of the manuscript, we have modified this part of the "Method" section (following also the suggestions of the first Referee) in order to better clarify our procedure in estimating the density of the new snow ($\rho_{new\,snow}$) used for deriving SWE from snow depth data (see answers to the main comments).

*Line 149: Why is this value presented as a new result in the abstract, if it has already been found in previous studies. But none of these studies really demonstrated how exactly the value was determined.*

Following also the suggestions of the first Referee, we have modified this paragraph and the Abstract. In the revised manuscript, we have explained better how the new snow density is estimated (see answers to the main comments).

*Lines 162-163: How does this compare with neighboring stations?*

We have added a comparison with data from 10 stations spread over the Italian Alps: "These values are in agreement with findings over the Italian Alps in the period 1960–2009. In fact, Valt and Cianfarra (2010) reported a mean snow depth of 233 cm (from 199 to 280 cm) for the stations above 1500 m a.s.l."

*Line 162:* We have modified the sentence accordingly adding "second".

*Line 169: You don't know without independent validation! Small snowfalls are impossible to detect due to uncertainty of the snow depth sensors and since there is often concurrent settling even snowfalls larger the measurement uncertainty can not be detected. impossible to detect and*

Following also the suggestions of the first Referee, we have deleted this paragraph regarding the comparison between Campbell and Sommer sensors.

*Lines 182-184:* We have deleted this sentence accordingly.

*Lines 186-187: This is strange! Could you provide any explanation? Why should the measurement be correct afterwards?*

The results from the snow pillow are difficult to explain as this sensor has been working for only two winter seasons and we are still in the process of testing. Analyzing data from the years to come will allow a more robust interpretation. However, we have searched for a possible explanation of this problem and this error could be due to the configuration of the snow pillow.

*Lines 191-192: Why do you think the snow pillow working correctly before January 2015, but not before January 2016?*

We found that the valid readings from the snow pillow occur whenever there is a snow cover thicker than about 50 cm. Because of the observed inter-annual variability in snow depth values, the beginning of the valid readings from the snow pillow differs from one year to the next. In fact, in the accumulation period 2015/2016 this threshold is reached after January 2016.

*Line 199: I suggest to add these measurements in Fig. 6.*

We have added in the Figure the values measured by means of the snow weighing tube, accordingly. The new version of the figure is:

[Figure]

*Line 205-206: Improve english!*

Following also the suggestion of the first Refeee, we have modified this paragraph.

*Line 206:* We have added "of the density by" accordingly.

*Lines 207-208: Percentage difference instead of the absolute values would provide much more information*.

Following also the suggestion of the first Referee, we have added "corresponding to about 17% of the mean total cumulative SWE considering all hydrological years".

*Line 212: really tens and not ten?*

The ice glacier surface features a lot of different conditions: bare ice, ponds of different size and depth, presence of dust and fine or coarse debris that can be scattered over the surface or aggregated. This surface heterogeneity translates into a differential ablation. In fact, each material has a different value of albedo and heat transfer and then the below ice will receive a different amount of energy for melting. The result is a surface roughness of tens of centimeters.

*Lines 223-224: Not clear what you mean?*

Following also the suggestions of the first Referee, we have completely modified this paragraph.

*Lines 227-229: Which would not be true in your case since you measure over ice!*

Following also the suggestions of the first Referee, we have completely modified this paragraph.

*Line 232-246: What is this? Does not belong to Discussion and has not direct connection to your results.*

Following also the suggestions of the first Referee, we have moved this part to the "Introduction" section.

*Lines 254-256: What is this temperature threshold for the AWS1?*

We have added accordingly "From the Forni Glacier, the application of the +0.5°C daily temperature threshold allows for a consistent quantification of snow ablation while, instead, for detecting the beginning of the snow melting processes, a suitable threshold has proven to be at least −4.6°C."

*Line 260:* Following also the suggestions of the first Referee, we have modified this part from "The unique issue is represented by the definition of…" to "Finally, the SR50 sonic ranger features the unique problem of the definition of the start of the accumulation period, but this can be overcome using albedo data."

*Figures:* We have modified all the figures and captions accordingly.

We very much appreciate the time and effort you put into the comments.

Sincerely,

Antonella Senese and Co-author

---

## Referee Report (RR1)

[referee-annotated manuscript omitted]

---

## Referee Report (RR2)

[referee-annotated manuscript omitted]

---

## Editor Decision (ED1)

**Editor comments on "Estimating the snow water equivalent on glacierized high elevation areas (Forni Glacier, Italy)" by Senese et al.**

**Abstract**

The second paragraph (starting with "The results indicate…" until "…, ranging from ±43 mm w.e. to ±144 mm w.e..") needs rewriting. This part is merely a listing of numbers but does not give enough context to a reader to understand if this manuscript is of interest for them or not.

Please define the abbreviation w.e. water equivalent in the manuscript similar to the explanation of SWE.

**Introduction and scientific background**

I think the term "new snow" and all related terms should be defined, that also opens for the suggestions of reviewer two to further explain the method used for estimating mean new snow density values and its limitations.

(page 2, line 41 – 52) The site description seems to be accidently squeezed into two paragraphs with scientific background and review of other studies. I suggest to locate the site description behind the scientific background and your defining of the research gaps, probably even with a separate subsection title – maybe under the data and methods chapter.

(page 2, line 59-66) I don't think it is necessary to describe the DFIR with so detailed wording, rather include a picture with the relevant citation and explain why a DFIR on your site (and other comparable sites) is not possible.

**Data and Methods**

(page 4, line 117) The coordinates of the site seem a bit out of contectx here. They would fit better in a separate site description section (if you decide for that, see comment above) or either before or behind the list of instruments instead of in between.

Consider a division into subsections to better guide the reader through the different topics you touch. Your chapter includes site description, instrument descriptions, challenges with measuring principles, some mechanical solutions, and mathematical methods rather mixed.

(page 5, line 176) You are talking about a strict control. I suggest to rather use the term "quality control" – if that is what you meant. Otherwise you may describe what do you mean by strict control? Was it manual, automatic? What are your thresholds, filter methods, …?

**Results**

(page 6, line 201) Please use the parentheses solely around the citation and include the number "equal to 140 kg/m3" into the sentence. Or rewrite, i.e. "The updated value of rho_newsnow is 149 kg/m3, which is similar to

(page 6, lines 206) I find it easy to misunderstand this sentence. It seems that you would need a period without data for not underestimating. Though I think, you want to say that from those periods with missing data, it becomes clear that the accumulation is underestimated and thus a complete dataset is very important? Consider rephrasing.

**Discussion**

I also suggest here some further division into subsections as you discuss a lot of topics. That eases the reading process and also helps you to organize your text in a more consequent manner.

(page 8, line 262). Please refer to the large deviations between the SWE values (independent of the chosen snow density) by the SR50 and the snow pit measurements in the years 2010, 2011, 2012 and 2013.

(page 8, line 276-280). Please find a more technical way to report your problems and possible solutions. Give the reader some trust that these are challenges your team can overcome.

(page 9, line 313). Please explain what you mean by inconsistent. Do you mean that the SR50 measure a random distance or the shortest distance or rather an average distance?

**Conclusions**

(page 11, line 401) – remove "for our limited experience in such remote areas" – I think you showed a good deal of experience with measurements in remote areas in your entire paper.

---

## Author Response (AR2)

Dear Editor,

We revised the manuscript in accordance with your comments and those from the reviewers. In particular, we have added further details regarding the definition of the new snow and we have discussed the settlement of the snow pack under the new snow layer.

We wish to thank you and the Reviewers for the helpful suggestions.

Please find below the detailed comments and responses to all suggestions. In the revised manuscript, the changes are colored in yellow.

Many thanks for your help.

Best regards,

Antonella Senese and co-authors

**REV #1**

*I think the authors did a good and relatively thorough job of correcting the deficiencies pointed out in the first round of review. There are still some minor revisions that are required before this manuscript can be published, including some modifications to improve clarity and interpretation of the results. My specific comments and suggestions are embedded in the attached revised manuscript in pdf format.*

We have addressed all the suggested corrections shown in the pdf file and in particular:

*L 20: Thank you for quantifying "good", however, reporting an RMSE here is ambiguous without reporting the mean maximum SWE for the site or the relative error. Can you do this as well?*
We have modified the sentence accordingly: "Once the new snow density is known, the sonic ranger allows deriving SWE values with a RMSE of 45 mm water equivalent (if compared with snow pillow measurements), that turns out to be about 8% of yearly average total SWE.".

*L 51: A reference would be good here:*
*Nitu, R., Rasmussen, R., Baker, B., Lanzinger, E., Joe, P., Yang, D., Smith, C., Roulet, Y., Goodison, B., Liang, H., Sabatini, F., Kochendorfer, J.,Wolff, M., Hendrikx, J., Vuerich, E., Lanza, L., Aulamo, O., and V. Vuglinsky: WMO intercomparison of instruments and methods for the measurement of solid precipitation and snow on the ground: organization of the experiment, WMO Technical Conference on meteorological and environmental instruments and methods of observations, Brussels, Belgium, https://www.wmo.int/pages/prog/www/IMOP/publications/IOM-109_TECO-2012/Session1/O1_01_Nitu_SPICE.pdf, 16–18, 2012.*
We have added the reference accordingly.

*L 52: This might be a good reference for this:*
*Key, J., Goodison, B., Schöner, W., Godøy, Ø, Ondráš, M., & Snorrason, Á.: A Global Cryosphere Watch, Arctic, 68, 48-58, 2015, http://www.jstor.org/stable/43871386*
We have added the reference accordingly.

*L 57: It's not quite clear what you mean by this sentence, reword. Perhaps this is better stated as: "For catchment type precipitation sensors, the catch efficiency of solid precipitation needs to be considered for the correct measurement of new snow."*
We have modified the sentence accordingly.

L 65: We have modified the sentence accordingly.

*L 66: this seems a bit out of context but I think you are trying to say that the installed instrumentation does not require wind shielding and therefore has substantially less infrastructure. Please clarify.*
We have modified the sentence accordingly: "For this reason, at the Forni Glacier, snow data have been acquired by means of sonic ranger and snow pillow instrumentations, without wind shielding."

*Line 67: this sentence is a bit confusing and doesn't really belong here. It perhaps belongs closer to the beginning where you justify the importance of snow measurements. More crucial here is the requirement*

*for a lead-in sentence for the importance of estimating new snow density when you only have snow depth measurements for estimating SWE.*

We have moved this sentence to the first paragraph accordingly. In addition, we have added a lead-in sentence accordingly: "For estimating SWE only from snow depth measurements, estimating a correct new snow density is crucial."

L 139: We have modified the sentence accordingly.

L 141: We have modified the sentence accordingly.

*L 146: This would read better as: "The automated instrument are sampled every 60 seconds. The SR50 sonic ranger, wind sensor samples...are averaged every 60-minutes. The air temperature, relative humidity...sample are averaged every 30-minutes." ...etc*

We have modified the sentences accordingly: "The automated instruments are sampled every 60 seconds. The SR50 sonic ranger, wind sensor and barometer samples are averaged every 60 minutes. The air temperature, relative humidity, solar and infrared radiation, and liquid precipitation sample are averaged every 30 minutes. The USH8 sonic ranger and snow pillow sample are averaged every 10 minutes. All data are recorded in a flash memory card, including the basic distribution parameters (minimum, mean, maximum, and standard deviation values).".

L 197: We have modified the sentence accordingly.

*L 204: This is better than how you stated it before but it's still hard to know if an RMSE of 58 mm w.e. is "relatively good" without at least reporting the mean of the intercomparison*

We have added accordingly: "with a mean $SWE_{snow-pit}$ value of 609 mm w.e.".

L 207: We have modified the sentence accordingly.

*L 217: I still have a problem with this sentence. It reads as if you were evaluating the snow pillow and not the SR50 technique. Which method are you evaluating? I believe, despite known problems with the snow pillow sensors, it would be the more proven technique and can be used to add credibility to your snow depth technique, not the other way around.*

We have modified this part accordingly: "Apart from a first period without or with a very thin cover of snow, the $SWE_{SR}$ curve follows the curve of SWE measured by the snow pillow (Fig. 4), thus suggesting that our approach seems to give reasonable results."

*L 223: again, reporting the maximum SWE for these two years would put these error numbers into context.*

We have modified accordingly: "Considering the whole dataset, the RMSE is 45 mm w.e., that turns out to be about 8% of yearly average total SWE measured by the snow pillow."

*L 248: Analysis shows*

We have modified the sentence accordingly.

L 250: We have modified the sentence accordingly.

*Line 262: This paragraph needs some revision. The paragraph starts by introducing errors associated with data gaps, then interjects a discussion about errors associated with large snowfall events with densities signficantly different from the average, then goes back to data gaps. Split up this discussion.*

*At my request, you added the discussion about the potential for large events with high densities to skew the SWE estimate. I think I'm satisfied with your answer but perhaps it's overexplained in the text. It might read better if you simply state that the technique is susceptible to this error, and then state that high precipitation amounts are infrequent (quoting the numbers), reducing the likelihood of this happening here. Without knowing the true density of the new snow during these big events, it's difficult to know their impact on the SWE estimate. As an estimation excercise, you could calculate the difference in SWE for a large event (e.g. 30 cm) if the new snow density is increased from 149 kg/m3 to 200 kg/m3.*

We have modified this paragraph accordingly: "In addition to an accurate definition of new snow density, an uninterrupted dataset of snow depth is also necessary in order to derive correct SWESR values. This can be deducted also observing the large deviations between the SWE values (independent of the chosen snow density) by the SR50 and the snow pit measurements in the years 2010, 2011, 2012 and 2013. It is therefore necessary to put in place all the available information to reduce the occurrence of data gaps to a minimum. The introduction of the second sonic ranger (Sommer USH8) at the end of the 2013-2014 snow season was an attempt to reduce the impact of this problem. This second sonic ranger, however, was still in the process of testing in the last years of the period investigated within this paper. We are confident that in the years to come it can help reduce the problem of missing data. Indeed, daily variations in snow depth measured by one sensor could be used to fill the data gap of the other one. Multiple sensors for fail-safe data collection are indeed highly recommended. In addition, the wooden four stakes installed at the corners of the snow pillow at the beginning of the 2014-2015 snow season were another idea for collecting more data. Unfortunately, they were broken almost immediately after the beginning of the snow accumulation period. They can be another way to deal with the problem of missing data, provided we figure out how to avoid breakage during the winter season. Probably the choice of a more robust and white material (such as insulated white steel) could overcome this issue.

It is also important to stress that potential errors in individual snowfall events could affect peak SWESR estimation. A large snowfall event with a considerable deviation from the mean new snow density will result in large errors (e.g. a heavy wet snowfall). These events are rather rare at the Forni site: only 3 days in the 11-year period covered by the data recorded more than 40 cm of new snow (the number of days decreases to 1 if the threshold increases to 50 cm). Therefore, even if the proposed technique can be susceptible to these errors, high precipitation amounts are infrequent, reducing the likelihood of this happening at the Forni site. Without knowing the true density of the new snow during these big events, it's difficult to know their impact on the SWE estimate. However, assuming that the new snow density could be increased from 149 kg m$^{-3}$ to 200 kg m$^{-3}$, the difference in SWE for a large event (e.g. 30 cm) is 15 mm w.e. (45 mm w.e. with 149 kg m$^{-3}$ and 60 mm w.e. with 200 kg m$^{-3}$)."

L 299: We have modified the sentence accordingly.

L 302: We have modified the sentence accordingly.

L 303: We have modified the sentence accordingly.

L 305: We have modified the sentence accordingly.

L 306: We have modified the sentence accordingly.

*L 315: This should be a new paragraph and I would suggest combining it with the discussion of other snow pillow errors that are included in the paragraph starting on line 328.*

We have modified these two paragraphs accordingly.

*L 342: it would be useful here to provide a rough percentage of how much of the difference in the intercomparison could be explained by spatial variability (i.e. up to 10%? 20%?)*
We have added accordingly: "the standard deviation is 54 mm w.e., corresponding to 12% of the mean value from snow weighing tube measurements".

L 347-351: We have deleted these sentences accordingly.

*L 371: I wouldn't say this since they are used at remote sites around the world.*
We have modified the sentence accordingly.

*L 389: Please add the relative error here as well.*
We have added accordingly: "(corresponding to 8% of the maximum SWE measured by means of the snow pillow)".

*L 396: I would add here: "...provided that the mean new snowfall density can be reliably estimated."*
We have added the sentence accordingly.

*L 397: I would re-phrase this such as: "Although conventional precipitation sensors, such as heating tipping bucket rain gauges, heated weighing gauges or disdrometers, can perhaps provide more accurate estimates of precipitation and SWE than the ones installed at Forni, they are less than ideal for use in high alpine and glacier sites." I would also implying that conventional sensors are more accurate...depending on the situation, that may not be the case.*
We have modified the sentence accordingly.

*REV 2*

*The paper definitively improved, but there is still a major fallacy in the study. The authors state "daily SR50 sonic ranger measures and the available snow pit data can be used to define the mean new snow density value at the site". This is wrong!*

*Correct is that the two instruments allow to determine a proxy for the new snow density. This proxy new snow density will always be higher than a measured new snow density, since the method totally neglects the settling of the old snow cover. This means that the measured daily positive snow depth difference is usually smaller than the real depth of the new snow and therefore explains the fact that your proxy of the mean new snow density is higher than any other mean new snow density reported in your introduction chapter.*

*I cannot accept the paper as long as the authors do not clearly state this fact throughout the paper.*

We understand the comment of the Reviewer 2 and we agree with the fact that settling processes of the snow pack under the new snow layer can occur at the Forni site, but this could affect our calculation mainly with snowfall lasting for several days. In this case, the measured daily positive snow depth differences could be smaller than the real depth of the new snow, with the consequence of overestimating new snow density. However, the obtained mean new snow density is not so higher than the general values found in literature as instead stated by the Referee. In the Introduction section, we reported the following values:

| Range | Mean | Reference |
|---|---|---|
| | 100 kg m$^{-3}$ | Roebber et al., 2003 * |
| 10-350 kg m$^{-3}$ | | Roebber et al., 2003 |
| 30-480 kg m$^{-3}$ | 123 kg m$^{-3}$ | Bocchiola and Rosso, 2007 (Central Italian Alps) |
| Min of 50 kg m$^{-3}$ | | Gray, 1979; Anderson and Crawford, 1990 |
| 10-257 kg m$^{-3}$ | 72-103 kg m$^{-3}$ | Judson and Doesken (2000) |
| | 120 or 200 kg m$^{-3}$ | Roebber et al. (2003) |
| 20-200 kg m$^{-3}$ | | Pahaut (1975) |

* Several authors stated that this value is an inadequate characterization of the true range of new snow densities (e.g. Currie, 1947; LaChapelle, 1962; Power et al., 1964; Super and Holroyd, 1997; Judson and Doesken, 2000).

In addition, the comparison with snow pillow dataset seems supporting our methodology. On the other hand, if many days pass between one snowfall and the following one, the settlement of the snow pack under the new snow layer is less likely to affect the measured differences in snow depth and this seems to be the case of the Forni Glacier site as snow days are only 9% of the snow season days.

Finally, this issue is one of the possible errors affecting our approach (please see section 5.1 Possible errors related to the methodology), but it is not the most crucial one. Therefore, we do not agree that our method is completely wrong: it represents the unique reliable approach that can be used in glacierized remote areas as demonstrated by our results.

We have further discussed about the settlement issue in the Discussion section accordingly: "Our new snow data could be affected by settling, sublimation, snow transported by wind, and rainfall. As far as settling is concerned, $\Delta h_{snow-pit-j}$ from Eq. 2 would indeed be higher if $\Delta h_{tj}$ values were calculated considering an interval shorter than 24 hours. However, this would not be possible because on the one hand, the sonic

ranger data's margin of error is too high to consider hourly resolution, and on the other hand, new snow is defined by the WMO within the context of a 24-hour period. Settling processes can concern also the snow pack under the new snow layer. This process can affect our daily differences mainly with snowfall lasting for several days. In this case, the measured daily positive snow depth differences could be smaller than the real depth of the new snow, with the consequence of overestimating new snow density. However, the obtained mean new snow density is not so higher than the general values found in literature. In addition, the comparison with snow pillow dataset seems supporting our methodology. On the other hand, if many days pass between one snowfall and the following one, the settlement of the snow pack under the new snow layer is less likely to affect the measured differences in snow depth and this seems to be the case of the Forni Glacier site as snow days are only 9% of the snow season days."

*A few minor remarks can be found in the attached PDF.*

We have addressed all the suggested corrections shown in the pdf file and in particular:

*Title: site (you don't do it for an area)*
We have modified the title accordingly.

*Abstract: The abstract repeats some statements and contains too much numbers!*
We have rewritten the Abstract section accordingly: "We present and compare 11 years of snow data (snow depth and snow water equivalent, SWE) measured by an Automatic Weather Station corroborated by data resulting from field campaigns on the Forni Glacier in Italy. The aim of the analyses is to estimate the SWE of new snowfall and the annual peak of SWE based on the average density of the new snow at the site (corresponding to the snowfall during the standard observation period of 24 hours) and automated snow depth measurements. The results indicate that the daily SR50 sonic ranger measures and the available snow pit data can be used to estimate the mean new snow density value at the site, with an error of $\pm 6$ kg m$^{-3}$. Once the new snow density is known, the sonic ranger allows deriving SWE values with a RMSE of 45 mm water equivalent (if compared with snow pillow measurements), that turns out to be about 8% of yearly average total SWE. Therefore, the methodology we present is interesting for remote locations such as glaciers or high alpine regions, as it allows estimating total snow water equivalent (SWE) using a relatively inexpensive, low power, low maintenance, and reliable instrument such as the sonic ranger."

L 18: We have added "snow" accordingly.

*L 19: "as well as to find the most appropriate method for evaluating SWE at this measuring site" ?*
We have deleted this part accordingly.

L 20-21: We have modified the sentence accordingly.

*Eq. 4: What means SR?*
SR means "Sonic Ranger". Then, we have added "(from sonic ranger data)" accordingly.

*L 285: Here you talk about the settling of new snow, but you totally neglect the settling of the old snow cover.*
As reported in the previous comment, we have added some comments regarding the settling processes that can occur during a 24-hour period: "Settling processes can concern also the snow pack under the new snow layer. This process can affect our daily differences mainly with snowfall lasting for several days. In this

case, the measured daily positive snow depth differences could be smaller than the real depth of the new snow, with the consequence of overestimating new snow density. However, the obtained mean new snow density is not so higher than the general values found in literature. In addition, the comparison with snow pillow dataset seems supporting our methodology. On the other hand, if many days pass between one snowfall and the following one, the settlement of the snow pack under the new snow layer is less likely to affect the measured differences in snow depth and this seems to be the case of the Forni Glacier site as snow days are only 9% of the snow season days."

*L 387: Please also add a relative measure.*
We have added the comparison with the maximum measured SWE accordingly: "(corresponding to 8% of the maximum SWE measured by means of the snow pillow)"

**EDITOR**

*Abstract*

*The second paragraph (starting with "The results indicate…" until "…, ranging from ±43 mm w.e. to ±144 mm w.e..") needs rewriting. This part is merely a listing of numbers but does not give enough context to a reader to understand if this manuscript is of interest for them or not.*
We have rewritten completely this part: "The results indicate that the daily SR50 sonic ranger measures and the available snow pit data can be used to estimate the mean new snow density value at the site, with an error of ±6 kg m$^{-3}$. Once the new snow density is known, the sonic ranger allows deriving SWE values with a RMSE of 45 mm water equivalent (if compared with snow pillow measurements), that turns out to be about 8% of yearly average total SWE. Therefore, the methodology we present is interesting for remote locations such as glaciers or high alpine regions, as it allows estimating total snow water equivalent (SWE) using a relatively inexpensive, low power, low maintenance, and reliable instrument such as the sonic ranger.".

*Please define the abbreviation w.e. water equivalent in the manuscript similar to the explanation of SWE.*
We have added "water equivalent" in the manuscript accordingly.

*Introduction and scientific background*

*I think the term "new snow" and all related terms should be defined, that also opens for the suggestions of reviewer two to further explain the method used for estimating mean new snow density values and its limitations.*
In the Introduction section, we already inserted the definition of new snow and snow depth: "The snow data thus acquired refer to snowfall or new snow (i.e. depth of freshly fallen snow deposited over a standard observation period, generally 24 hours, see WMO, 2008; Fierz et al., 2009) and to snow depth (i.e. the total depth of snow on the ground at the time of observation, see WMO, 2008).". In addition, in the Abstract we already specify the period considered for the new snow: "The aim of the analyses is to estimate the SWE of new snowfall and the annual peak of SWE based on the average density of the new snow at the site (corresponding to the snowfall during the standard observation period of 24 hours) and automated snow depth measurements, as well as to find the most appropriate method for evaluating SWE at this measuring site."
In order to better clarify our approach, we have added the definition of new snow also in the Data and method section.

*(page 2, line 41 – 52) The site description seems to be accidently squeezed into two paragraphs with scientific background and review of other studies. I suggest to locate the site description behind the scientific background and your defining of the research gaps, probably even with a separate subsection title – maybe under the data and methods chapter.*
We have added a new section (2 Study area and Forni AWSs) between 1 Introduction and 3 Data and methods:

[revised manuscript text omitted]

*(page 2, line 59-66) I don't think it is necessary to describe the DFIR with so detailed wording, rather include a picture with the relevant citation and explain why a DFIR on your site (and other comparable sites) is not possible.*
We have modified the sentence accordingly, deleting the detailed explanation of the DFIR.

**Data and Methods**

*(page 4, line 117) The coordinates of the site seem a bit out of contectx here. They would fit better in a separate site description section (if you decide for that, see comment above) or either before or behind the list of instruments instead of in between.*
We have added a new section (2 Study area and Forni AWSs) regarding the study area in which we have inserted the coordinates of the site.

*Consider a division into subsections to better guide the reader through the different topics you touch. Your chapter includes site description, instrument descriptions, challenges with measuring principles, some mechanical solutions, and mathematical methods rather mixed.*
We have added a new section (2 Study area and Forni AWSs) between 1 Introduction and 3 Data and methods, in which we have focused only on the site and AWS description. The Data and methods section now regards only snow data and methods for estimating new snow density and SWE.

*(page 5, line 176) You are talking about a strict control. I suggest to rather use the term "quality control" – if that is what you meant. Otherwise you may describe what do you mean by strict control? Was it manual, automatic? What are your thresholds, filter methods, …?*
We have modified the sentence accordingly adding the term "quality".

**Results**

*(page 6, line 201) Please use the parentheses solely around the citation and include the number "equal to 140 kg/m3" into the sentence. Or rewrite, i.e. "The updated value of rho_newsnow is 149 kg/m3, which is similar to*
We have modified the sentence accordingly.

*(page 6, lines 206) I find it easy to misunderstand this sentence. It seems that you would need a period without data for not underestimating. Though I think, you want to say that from those periods with missing data, it becomes clear that the accumulation is underestimated and thus a complete dataset is very important? Consider rephrasing.*
We have modified the sentence accordingly : "In particular, in addition to the length of missing dataset, the period of the year with missing data influences the magnitude of the actual accumulation underestimation."

*Discussion*

*I also suggest here some further division into subsections as you discuss a lot of topics. That eases the reading process and also helps you to organize your text in a more consequent manner.*
We have divided the Discussion section in two parts: 5.1 Possible errors related to the methodology and 5.2 Possible errors related to the instrumentation.

*(page 8, line 262). Please refer to the large deviations between the SWE values (independent of the chosen snow density) by the SR50 and the snow pit measurements in the years 2010, 2011, 2012 and 2013.*
We have added this comparison accordingly.

*(page 8, line 276-280). Please find a more technical way to report your problems and possible solutions. Give the reader some trust that these are challenges your team can overcome.*
We have added some sentences accordingly: "We are confident that in the years to come it can help reduce the problem of missing data. Indeed, daily variations in snow depth measured by one sensor could be used to fill the data gap of the other one. Multiple sensors for fail-safe data collection are indeed highly recommended. In addition, the wooden four stakes installed at the corners of the snow pillow at the beginning of the 2014-2015 snow season were another idea for collecting more data. Unfortunately, they were broken almost immediately after the beginning of the snow accumulation period. They can be another way to deal with the problem of missing data, provided we figure out how to avoid breakage during the winter season. Probably the choice of a more robust and white material (such as insulated white steel) could overcome this issue."

*(page 9, line 313). Please explain what you mean by inconsistent. Do you mean that the SR50 measure a random distance or the shortest distance or rather an average distance?*
We have added accordingly: "generally much smaller than the values of the previous and subsequent readings".

*Conclusions*

*(page 11, line 401) – remove "for our limited experience in such remote areas" – I think you showed a good deal of experience with measurements in remote areas in your entire paper*
We have deleted this part accordingly.

---

## Author Response (AR3)

Dear Editor,

We revised the manuscript in accordance with your suggestion and then the standard of English spelling and grammar has been improved by a professional, mother-tongue consultant.

In the revised manuscript, the changes are colored in yellow.

Many thanks for your help.

Best regards,

Antonella Senese and co-authors